# RECTIFIED DIFFUSION: STRAIGHTNESS IS NOT YOUR NEED IN RECTIFIED FLOW

**Fu-Yun Wang**[1]  **Ling Yang**[2]  **Zhaoyang Huang**[1]  **Mengdi Wang**[3]  **Hongsheng Li**[1,4]

[1] MMLab, CUHK, Hong Kong SAR  [2] Peking University, Beijing, China
[3] Princeton University, New Jersey, USA  [4] CPII under InnoHK, Hong Kong SAR
`fywang@link.cuhk.edu.hk, yangling0818@163.com, drinkingcoder@link.cuhk.edu.hk`
`mengdiw@princeton.edu, hsli@ee.cuhk.edu.hk`

## ABSTRACT

Diffusion models have greatly improved visual generation but are hindered by slow generation speed due to the computationally intensive nature of solving generative ODEs. Rectified flow, a widely recognized solution, improves generation speed by straightening the ODE path. Its key components include: 1) using the linear interpolating diffusion form of flow-matching, 2) employing $v$-prediction, and 3) performing rectification (a.k.a. reflow). In this paper, we argue that the success of rectification primarily lies in using a pretrained diffusion model to obtain matched pairs of noise and samples, followed by retraining with these matched noise-sample pairs. Based on this, components 1) and 2) are unnecessary. Furthermore, we highlight that straightness is not an essential training target for rectification; rather, it is a specific case of flow-matching models. The more critical training target is to achieve a first-order approximate ODE path, which is inherently curved for models like DDPM and Sub-VP. Building on this insight, we propose Rectified Diffusion, which generalizes the design space and application scope of rectification to encompass the broader category of diffusion models, rather than being restricted to flow-matching models. We validate our method on Stable Diffusion v1-5 and Stable Diffusion XL. Our method not only greatly simplifies the training procedure of rectified flow-based previous works (e.g., InstaFlow) but also achieves superior performance with even lower training cost. Our code is available at `https://github.com/G-U-N/Rectified-Diffusion`.

## 1 INTRODUCTION

Diffusion models have greatly advanced the field of visual generation, enabling the creation of high-quality images and vivid videos from text (Ho et al., 2020; Song et al., 2020b; Rombach et al., 2022a; Singer et al., 2022; Podell et al., 2023; Esser et al., 2024; Shi et al., 2024). However, the generation process of diffusion models involves solving an expensive generative ODE numerically, which significantly slows down the generation speed compared to other generative models (e.g., GAN) (Goodfellow et al., 2020; Sauer et al., 2023b;a). A widely recognized solution to this issue is rectified flow. The training target of rectified flow, as highlighted in the previous works (Liu et al., 2023; 2022; Yan et al., 2024), is to make the new ODE path straighter, enabling the models to generate high-fidelity images with fewer steps while retaining the flexibility of sampling with more inference steps for further quality enhancement. The key components of rectified flow are threefold:

1) **Flow-Matching.** Rectified flow proposes to employ the linear interpolating diffusion form (Liu et al., 2022; Lipman et al., 2022), which is also known as flow matching form. The intermediate noisy state $\mathbf{x}_t$ is defined as $(1-t)\mathbf{x}_0 + t\boldsymbol{\epsilon}$, where $\mathbf{x}_0$ is the clean data, $\boldsymbol{\epsilon} \sim \mathcal{N}(\mathbf{0}, \mathbf{I})$ is normal noise, and $t \in [0, 1]$ is the timestep. This design is more straightforward compared to the semi-linear form of the original DDPM (Ho et al., 2020).

2) **$v$-prediction.** Rectified flow proposes to adopt $v$-prediction (Salimans & Ho, 2022; Liu et al., 2022). That is, the model learns to predict $v = \mathbf{x}_0 - \boldsymbol{\epsilon}$. This makes the denoising form simple. For example, one can predict $\mathbf{x}_0$ based on $\mathbf{x}_t$ with $\hat{\mathbf{x}}_0 = \mathbf{x}_t + t\hat{v}_{\boldsymbol{\theta}}$, where $\boldsymbol{\theta}$ denotes the model

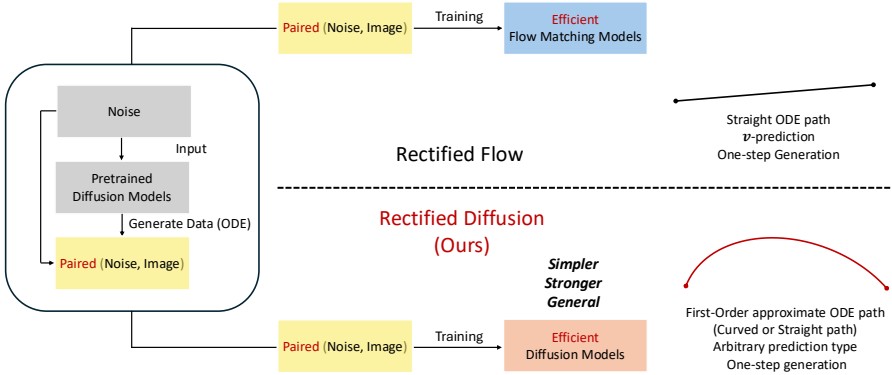

Figure 1: Overview of comparison between rectified flow and our rectified diffusion.

parameters and ˆ denotes the predictions. Moreover, it avoids the numerical issue when $t \approx 1$ with $\epsilon$-prediction. For example, $\hat{\mathbf{x}}_0 = \frac{\mathbf{x}_t - t\hat{\epsilon}_{\theta}}{1-t} \approx \frac{\mathbf{x}_t - t\hat{\epsilon}_{\theta}}{0}$, which is invalid.

3) **Rectification.** Rectification, also known as reflow, is an important technique proposed in rectified flow (Liu et al., 2022). It is a progressive retraining technique that greatly improves the generation quality at low-step regime and maintains the flexibility of standard diffusion models. To be specific, it turns an arbitrary coupling of $\mathbf{x}_0 \sim \mathbb{P}_0$ (real data) and $\epsilon \sim \mathbb{P}_1$ (noise) adopted in standard diffusion training to a new deterministic coupling of $\hat{\mathbf{x}}_0 \sim \mathbb{P}_0^{\theta}$ (generated data) and $\epsilon \sim \mathbb{P}_1$ (pre-collected noise). To put it in a nutshell, it replaces the $\mathbf{x}_t = (1-t)\mathbf{x}_0 + t\epsilon$ with $\mathbf{x}_t = (1-t)\hat{\mathbf{x}}_0 + t\hat{\epsilon}$, where $\mathbf{x}_0$ is real data, $\hat{\mathbf{x}}_0$ is data generated by pretrained diffusion models $\theta$, $\epsilon$ is the randomly sampled noise, and $\hat{\epsilon}$ is the noise used to generate data $\hat{\mathbf{x}}_0$. Previous works emphasize the rectification procedure is only feasible to $\mathbf{v}$-prediction based flow-matching models. That is, they believe the first two points are the foundations to adopt rectification for improving efficiency. And they emphasize the rectification procedure 'straightens' the ODE path.

The motivation of this paper is to *investigate what is most essential about rectified flow.* We argue that the effectiveness of rectified flow stems from using a pretrained diffusion model to acquire matched pairs of noise and samples, followed by retraining with these matched noise-sample pairs (i.e., the aforementioned third point). Based on this, the aforementioned first two points (i.e., flow-matching & $\mathbf{v}$-prediction) are unnecessary. This allows us to generalize the design space of rectified flow and make it adaptable to various diffusion variants, including DDPM (Ho et al., 2020), EDM (Karras et al., 2022), Sub-VP (Song et al., 2020b), and etc.

To this end, we propose rectified diffusion, as illustrated in Fig. 1, our overall design is straightforward. We keep everything of the pretrained diffusion models unchanged, including noise schedulers, prediction types, network architectures, and even training and inference code. The only difference is that the noise $\epsilon$ and data $\mathbf{x}_0$ adopted for training are pre-collected and generated by the pretrained diffusion models instead of independently sampled from Gaussian and real data datasets.

Additionally, *we highlight that straightness is no longer an essential training target* when we generalize the design space from solely linear interpolating diffusion form to more general diffusion forms. We analyze and show that the training target of rectified diffusion is to obtain a *first-order approximate ODE path*[1]. In simple terms, a first-order approximate ODE path implies the predictions of models remain consistent along the ODE trajectory and it still maintains at the same ODE trajectory after each denoising step. For models like DDPM (Ho et al., 2020), the first-order approximate ODE path is inherently curved instead of straight. Therefore, 'straightness' is no longer suitable for rectified diffusion and is just a special case when we use the form of flow-matching.

---

[1]Note: It should be clarified that the "first-order approximate ODE path" in this paper is distinct from the traditional concept of a "first-order ODE". In the conventional sense, a first-order ODE does not include terms such as $\frac{d^n x}{dt^n}$ where $n > 1$. In this context, the ODE of general diffusion models is already "first-order ODE". However, in our work, the "first-order approximate path" refers to an ODE with the property of being immune to high-order discretization errors. This distinction will become clearer in later sections.

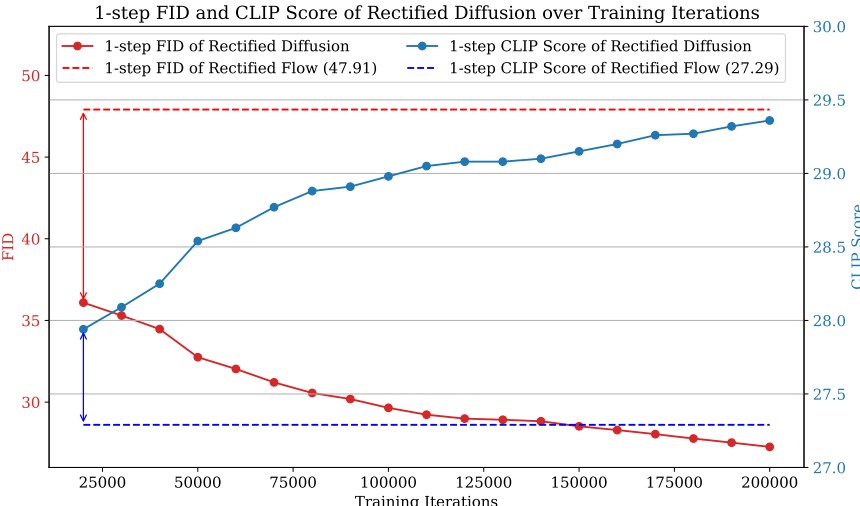

Figure 2: Training iterations. 1-step performance of rectified diffusion significantly surpasses the 1-step performance of rectified flow within only 20,000 iterations with batch size 128 (**only 8%** trained images of rectified flow) and consistently grows with more training iterations. The dashed lines represent the final performance of the rectified flow (Liu et al., 2023). Since the training code of rectified flow (Liu et al., 2023) is not open-sourced, intermediate metrics cannot be tested.

To empirically validate our claim, we conduct experiments using Stable Diffusion, comparing our approach with InstaFlow (Liu et al., 2023), a key baseline based on rectified flow for text-to-image generation. We adhere to the training setting of InstaFlow. The primary distinction is that InstaFlow requires transforming the Stable Diffusion into a $v$-prediction flow-matching model, while our method leaves everything of the original Stable Diffusion unchanged. Our results demonstrate apparently better performance and faster training, likely due to our minimal differences in diffusion configurations. Our one-step performance achieves significantly superior performance with only 8% trained images of InstaFlow as shown in Fig. 2.

Besides, we propose to replace the second-stage distillation adopted in InstaFlow with consistency distillation. We observe that the first-order approximate ODE path greatly facilitates consistency distillation, allowing us to achieve better performance at 3% the GPU days than the further distilled model of InstaFlow. Additionally, we introduce rectified diffusion (Phased), which divides the ODE path along the time axis into multiple segments and enforces first-order approximate ODE paths within each segment. While this segmentation increases the minimum number of generation steps to match the number of segments, it substantially reduces both training costs and time. When compared to the previous segment-based rectified flow method, PeRFlow (Yan et al., 2024), our approach demonstrates significantly better performance in experiments conducted on Stable Diffusion v1-5 (Rombach et al., 2022a) and Stable Diffusion XL (Podell et al., 2023).

We summarize our main contributions as follows: **(i)** We conduct an in-depth analysis of the essence of rectification and extend rectified flow to rectified diffusion. **(ii)** We identify that it is not straightness but first-order approximate ODE path is the essential training target of rectified diffusion. **(iii)** Comprehensive comparisons on rectification, distillation, and phased OED segmentation demonstrate our method achieves superior trade-off between generation quality and training efficiency over rectified flow-based models.

## 2 RECTIFIED DIFFUSION: GENERALIZING THE DESIGN SPACE OF RECTIFIED FLOW INTO GENERAL DIFFUSION MODELS

**Rectified flow is a subset of rectified diffusion.** In the following discussion, we apply the general diffusion form (Kingma et al., 2021) $\mathbf{x}_t = \alpha_t \mathbf{x}_0 + \sigma_t \boldsymbol{\epsilon}$ to introduce rectified diffusion. Note that this form of diffusion covers the flow-matching since we can simply set $\alpha_t = 1 - t$ and $\sigma_t = t$. Con-

sidering the different prediction types, we apply the epsilon-prediction for the following discussion. But note that different prediction types can be converted effortlessly through re-parameterization. For $\mathbf{x}_0$-prediction, we have $\mathbf{x}_0 = \frac{\mathbf{x}_t - \sigma_t \boldsymbol{\epsilon}}{\alpha_t}$. For $\boldsymbol{v}$-prediction utilized in rectified flow, we have $\boldsymbol{v} = \mathbf{x}_0 - \boldsymbol{\epsilon} = \frac{\mathbf{x}_t - (\alpha_t + \sigma_t)\boldsymbol{\epsilon}}{\alpha_t}$. Hence, we claim that *rectified flow is a subset of rectified diffusion, and rectified diffusion is a generalization of rectified flow.*

## 2.1 THE NATURE OF RECTIFICATION IS THE RETRAINING WITH PRE-COMPUTED NOISE-SAMPLE PAIR

**The secret of rectification is using paired noise-sample for training.** To illustrate the differences clearly, we visualize the training processes for standard flow matching and rectification (reflow) training, as described in Algorithm 1 and Algorithm 3, respectively. Differences are highlighted in red. A key observation is that in standard flow matching training, $\mathbf{x}_0$ represents real data randomly sampled from the training set, while the noise $\boldsymbol{\epsilon}$ is also randomly sampled from Gaussian. This results in random pairing between noise and sample. In contrast, in rectification training, the noise is pre-sampled from Gaussian, and the images are generated using pre-sampled noise by the model from the previous round of rectification (the pre-trained model), leading to a deterministic pairing.

**Flow-matching training (with linear interpolating form) is a subset of standard diffusion training.** In addition, Algorithm 2 visualizes the training process of a more general diffusion model, with differences to Algorithm 1 highlighted in blue and orange. It's important to note that flow matching is a specific case of the diffusion forms we discuss. From the algorithms, it is evident that the only distinctions between them lie in the diffusion form and prediction type. Consequently, flow matching training with linear interpolating form is just a special case of general diffusion training under a particular diffusion form and prediction type.

By comparing Algorithms 2 and 3 with Algorithm 1, it is straightforward to derive Algorithm 4. Essentially, by incorporating the pre-trained model to collect noise-sample pairs and replacing the randomly sampled noise and real samples with these pre-collected pairs in the general diffusion training, we obtain the training algorithm for rectified diffusion.

---

**Algorithm 1** Flow Matching $\boldsymbol{v}$-Prediction

**Input:**
Sample $\mathbf{x}_0$ from the data distribution
Sample time $t$ from a predefined schedule or uniformly from $[0, 1]$
Sample noise $\boldsymbol{\epsilon}$ from normal distribution
Compute $\mathbf{x}_t : \mathbf{x}_t = (1 - t) \cdot \mathbf{x}_0 + t \cdot \boldsymbol{\epsilon}$
Predict velocity $\hat{\boldsymbol{v}}$ using the model: $\hat{\boldsymbol{v}} = \text{Model}(\mathbf{x}_t, t)$
Compute loss: $\mathcal{L} = \|\hat{\boldsymbol{v}} - (\mathbf{x}_0 - \boldsymbol{\epsilon})\|_2^2$
Backpropagate and update parameters

---

**Algorithm 2** Diffusion Training $\boldsymbol{\epsilon}$-Prediction

**Input:** $\alpha_t$, $\sigma_t$
Sample $\mathbf{x}_0$ from the data distribution
Sample time $t$ from a predefined schedule or uniformly from $[0, 1]$
Sample noise $\boldsymbol{\epsilon}$ from normal distribution
Compute $\mathbf{x}_t : \mathbf{x}_t = \alpha_t \cdot \mathbf{x}_0 + \sigma_t \cdot \boldsymbol{\epsilon}$
Predict noise $\hat{\boldsymbol{\epsilon}}$ using the model: $\hat{\boldsymbol{\epsilon}} = \text{Model}(\mathbf{x}_t, t)$
Compute loss: $\mathcal{L} = \|\hat{\boldsymbol{\epsilon}} - \boldsymbol{\epsilon}\|_2^2$
Backpropagate and update parameters

---

**Algorithm 3** Rectified Flow $\boldsymbol{v}$-Prediction

**Input:** noise-data pair $(\boldsymbol{\epsilon}, \hat{\mathbf{x}}_0)$
~~Sample $\mathbf{x}_0$ from the data distribution~~
Sample time $t$ from a predefined schedule or uniformly from $[0, 1]$
~~Sample noise $\boldsymbol{\epsilon}$ from normal distribution~~
Compute $\mathbf{x}_t : \mathbf{x}_t = (1 - t) \cdot \hat{\mathbf{x}}_0 + t \cdot \boldsymbol{\epsilon}$
Predict velocity $\hat{\boldsymbol{v}}$ using the model: $\hat{\boldsymbol{v}} = \text{Model}(\mathbf{x}_t, t)$
Compute loss: $\mathcal{L} = \|\hat{\boldsymbol{v}} - (\hat{\mathbf{x}}_0 - \boldsymbol{\epsilon})\|_2^2$
Backpropagate and update parameters

---

**Algorithm 4** Rectified Diffusion $\boldsymbol{\epsilon}$-Prediction

**Input:** noise-data pair $(\boldsymbol{\epsilon}, \hat{\mathbf{x}}_0)$, $\alpha_t$, $\sigma_t$
~~Sample $\mathbf{x}_0$ from the data distribution~~
Sample time $t$ from a predefined schedule or uniformly from $[0, 1]$
~~Sample noise $\boldsymbol{\epsilon}$ from normal distribution~~
Compute $\mathbf{x}_t : \mathbf{x}_t = \alpha_t \cdot \hat{\mathbf{x}}_0 + \sigma_t \cdot \boldsymbol{\epsilon}$
Predict noise $\hat{\boldsymbol{\epsilon}}$ using the model: $\hat{\boldsymbol{\epsilon}} = \text{Model}(\mathbf{x}_t, t)$
Compute loss: $\mathcal{L} = \|\hat{\boldsymbol{\epsilon}} - \boldsymbol{\epsilon}\|_2^2$
Backpropagate and update parameters

---

## 2.2 UNDERSTANDING THE FIRST-ORDER APPROXIMATE ODE PATH (⋆⋆⋆)

**How do we define a first-order approximate ODE path, and what are its requirements?** For the above discussed general diffusion form $\mathbf{x}_t = \alpha_t \mathbf{x}_0 + \sigma_t \boldsymbol{\epsilon}$, there exists an exact ODE solution form (Lu et al., 2022),

$$\mathbf{x}_t = \frac{\alpha_t}{\alpha_s} \mathbf{x}_s - \alpha_t \int_{\lambda_s}^{\lambda_t} e^{-\lambda} \boldsymbol{\epsilon}_{\boldsymbol{\theta}}(\mathbf{x}_{t_\lambda}, t_\lambda) \mathrm{d}\lambda , \tag{1}$$

where $\lambda_t = \ln \frac{\alpha_t}{\sigma_t}$, and $t_\lambda$ is the inverse function of $\lambda_t$. As indicated in previous work (Kingma et al., 2021), $\lambda_t$ should be a monotonically decreasing function. The left term $\frac{\alpha_t}{\alpha_s} \mathbf{x}_s$ is a pre-defined deterministic scaling. The right term is the exponentially weighted integral of epsilon predictions. The first-order approximate ODE path means the above integral with arbitrary $t$ and $s$ is equivalent to

$$\mathbf{x}_t = \frac{\alpha_t}{\alpha_s} \mathbf{x}_s - \alpha_t \boldsymbol{\epsilon}_{\boldsymbol{\theta}}(\mathbf{x}_s, s) \int_{\lambda_s}^{\lambda_t} e^{-\lambda} \mathrm{d}\lambda = \frac{\alpha_t}{\alpha_s} \mathbf{x}_s + \alpha_t \boldsymbol{\epsilon}_{\boldsymbol{\theta}}(\mathbf{x}_s, s)(\frac{\sigma_t}{\alpha_t} - \frac{\sigma_s}{\alpha_s}) . \tag{2}$$

We show that the equivalent of Equation 1 and Equation 2 for arbitrary $t$ and $s$ *holds and only holds* if the epsilon prediction on the same ODE trajectory is a constant in Thereom 1.

**What is the principle behind the first-order approximate ODE path?** The equivalence between Equation 1 and Equation 2 does not generally hold across most cases. Specifically, Equation 2 serves as a first-order Taylor series approximation of Equation 1. Let's assume the outcome of Equation 1 is represented as $f(s,t)$ and that of Equation 2 as $\tilde{f}(s,t)$. Their relationship can then be written as:

$$f(s,t) = \tilde{f}(s,t) + \mathcal{O}((\lambda_t - \lambda_s)^2) . \tag{3}$$

Widely adopted ODE solvers in the community, such as DDIM and the Euler solver, operate under the premise that Equation 1 and Equation 2 are interchangeable, thus disregarding the higher-order error terms. These terms, denoted as $\mathcal{O}((\lambda_s - \lambda_t)^2)$, are typically overlooked during inference, leading to issues when the gap $\lambda_s - \lambda_t$ grows significant. This explains why DDIM necessitates a minimum of 50 steps to produce high-quality results. By contrast, our work sidesteps such approximations. Our key aim is to enhance the PF-ODE to guarantee first-order accuracy, eliminating higher-order error terms via a paired retraining approach (rectification). This underpins why Rectified Diffusion achieves high-quality generation in a single step.

**Why rectified diffusion leads to a first-order approximate ODE path?** Suppose the ODE trajectory is already a first-order approximate ODE path with a solution point $\mathbf{x}_0$. We define the constant epsilon prediction along this ODE trajectory as $\boldsymbol{\epsilon}$. Substituting $s = 0$, $\mathbf{x}_0$, $\alpha_s = 1$, $\sigma_s = 0$, and $\boldsymbol{\epsilon}$ into Equation 2, we obtain:

$$\mathbf{x}_t = \frac{\alpha_t}{1} \mathbf{x}_0 + \alpha_t \boldsymbol{\epsilon} \left( \frac{\sigma_t}{\alpha_t} - \frac{0}{1} \right) = \alpha_t \mathbf{x}_0 + \sigma_t \boldsymbol{\epsilon}. \tag{4}$$

This matches the structure of the predefined forward process precisely. Hence, the first-order approximate ODE path represents a weighted blend of data and noise, aligning with the predefined forward diffusion framework. The difference, however, is that in this equation, $\boldsymbol{\epsilon}$ and $\mathbf{x}_0$ are a deterministic pair tied to the same ODE trajectory, while in standard diffusion training, $\mathbf{x}_0$ and $\boldsymbol{\epsilon}$ are sampled randomly.

Now, if we achieve perfect coupling between the noise $\boldsymbol{\epsilon}$ and data $\mathbf{x}_0$ during training and ensure the no-intersection condition holds, $\mathbf{x}_t$ will correspond to a unique noise-data pair. In the absence of optimization errors, this leads to:

$$\boldsymbol{s}_{\boldsymbol{\theta}}(\mathbf{x}_t, t) = \nabla_{\mathbf{x}_t} \log \mathbb{P}(\mathbf{x}_t \mid \mathbf{x}_0) = -\frac{\mathbf{x}_t - \alpha_t \mathbf{x}_0}{\sigma_t^2} = -\frac{\boldsymbol{\epsilon}}{\sigma_t}. \tag{5}$$

Given that $\boldsymbol{\epsilon}_{\boldsymbol{\theta}} = -\sigma_t \boldsymbol{s}_{\boldsymbol{\theta}}$, it follows that the epsilon prediction along the PF-ODE will consistently equal $\boldsymbol{\epsilon}$. Referring back to Theorem 1, this implies that we successfully recover the first-order approximate ODE path.

This indicates that, with perfect pairing of the data $\mathbf{x}_0$ and noise $\boldsymbol{\epsilon}$ during training, and provided there are no overlaps between distinct paths (which could otherwise result in epsilon predictions averaging

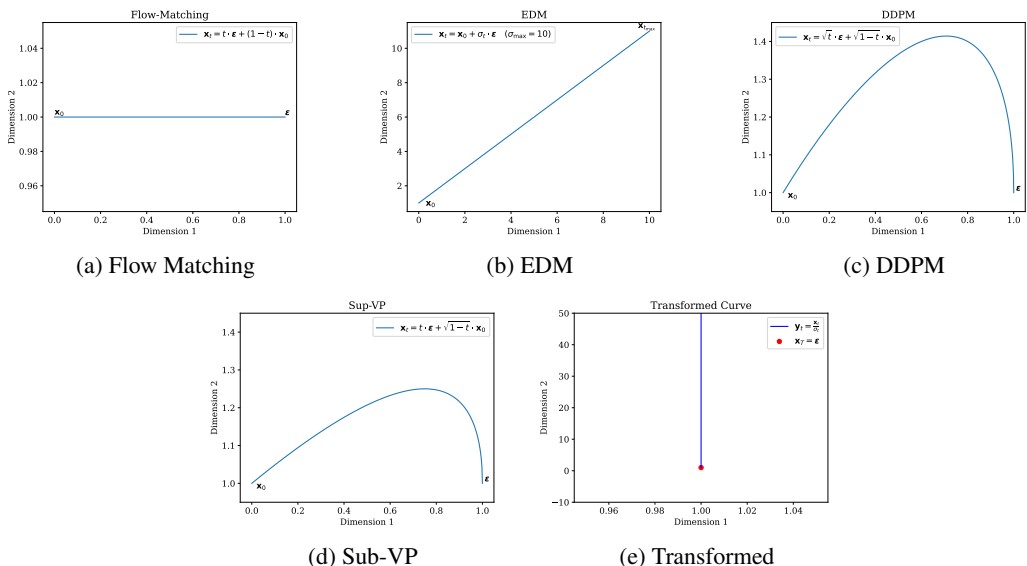

Figure 3: First-order approximate ODE paths of different diffusion forms. We show that the first-order approximate ODE path has the same form as their predefined forward process, i.e., $\mathbf{x}_t = \alpha_t \mathbf{x}_0 + \sigma_t \epsilon$. Though the first-order approximate ODE paths of Flow Matching and EDM are straight, the first-order approximate ODE paths of DDPM and Sub-VP are inherently curved. First-order approximate ODE paths of all diffusion forms can be converted into straight lines through simple scaling as shown in Fig. 3e.

over multiple paths), a diffusion model trained under ideal conditions (without optimization errors) would yield the first-order approximate ODE path. Indeed, as indicated by previous work (Lee et al., 2024), the probability of overlaps between distinct paths is actually close to zero with only once rectification. Additional insights are provided in the supplementary Section IV.

**First-order approximate ODE path supports consistent generation with arbitrary inference steps.** Additionally, note that if the epsilon predictions on the same trajectory are constant, it is easy to show that the $\mathbf{x}_0$-predictions are also constant. Therefore, the first-order approximate ODE path can flexibly support one-step generation ($\mathbf{x}_T \rightarrow \mathbf{x}_0$) or multi-step generation ($\mathbf{x}_T \rightarrow \cdots \rightarrow \mathbf{x}_0$). If a perfect first-order approximate ODE path is achieved, we will always get identical generation results with arbitrary inference steps.

**First-order approximate ODE path can be inherently curved.** For the first-order approximate ODE path, though the trajectories of flow-matching based methods are straight, the trajectories of other forms of diffusion models can be inherently curved. We showcase our findings in Fig. 3. We select $\mathbf{x}_0 = [0, 1]$ and $\epsilon = [1, 1]$ The Fig. 3a and Fig. 3b show the first-order approximate ODE paths of flow-matching and EDM. They are both straight, but EDM has a totally different trajectory and magnitude. Fig. 3c and Fig. 3d show the first-order approximate ODE paths of DDPM and Sub-VP. Their first-order approximate ODE paths are inherently curved. But if we define $\mathbf{y}_t = \frac{\mathbf{x}_t}{\sigma_t}$, we will have $\mathbf{y}_t = \frac{\alpha_t}{\sigma_t} \mathbf{x}_0 + \epsilon$ from the Equation 4. We can easily observe that the trajectory of $\mathbf{y}_t$ is a straight line from the initial point $\epsilon$ towards the direction of $\mathbf{x}_0$ (i.e, first-order trajectories can be converted to straight lines). Fig. 3e shows the trajectory of $\mathbf{y}_t = \frac{\alpha_t}{\sigma_t} \mathbf{x}_0 + \epsilon$. It shows that all the first-order trajectories can be converted into straight lines with simple timestep-dependent scaling.

## 2.3 RECTIFIED DIFFUSION (PHASED)

Completely smoothen the ODE path of a pre-trained diffusion model into first-order approximate ODE path is challenging because the original ODE can deviate significantly from the target ODE. In Fig. 4, we visualize both the original diffusion ODE path and the corresponding first-order approximate ODE path. *Since it's hard to intuitively determine whether a curved ODE path satisfies the property, we represent the first-order approximate ODE path with a straight line.* A significant gap between the two paths is evident. However, enforcing local first-order approximate ODE path is more feasible. As shown on the right side of the figure, when the ODE path is divided into two seg-

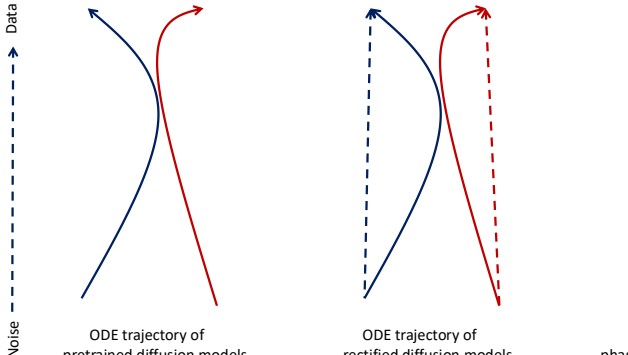

Figure 4: ODE trajectory comparison of diffusion models, rectified diffusion models, and phased consistency models. Since it's hard to visually tell whether a curved ODE path satisfies first-order property, *we apply straight lines for more clear demonstration.* The solid line shows the original diffusion ODE path, while the dashed line shows the rectified ODE path.

ments along the time axis and each segment is smoothened separately, the new ODE path is closer to the original one. This observation motivates the development of our rectification diffusion (phased).

We set intermediate time steps as $s_0 = 0 < s_1 < s_2 < \cdots < s_{M-1} = t_{\max}$ along the time axis of ODE, where $M$ is the number of phases. The training process begins with sampling $\mathbf{x}_0$ from real data, followed by adding random noise at time step $s_m$ to obtain $\mathbf{x}_{s_m}$. We then use the pre-trained diffusion model to perform multi-step numerical solving to obtain $\mathbf{x}_{s_{m-1}}$ for the previous intermediate step. However, the phasing idea involves two challenges: 1) determining the noise $\boldsymbol{\epsilon}$ corresponding to the first-order approximate ODE path, and 2) determine the sample $\mathbf{x}_t$ at any time $t$ between $s_m$ and $s_{m-1}$ on the same first-order approximate ODE path, where $t \in (s_{m-1}, s_m)$. Fortunately, the transition formula between any two points on the first-order approximate ODE path is known, as shown in Equation 2. Through a simple transformation, we have the noise $\boldsymbol{\epsilon}$ corresponding to the ODE path between $\mathbf{x}_{s_m}$ and $\mathbf{x}_{s_{m-1}}$ can be expressed as:

$$\boldsymbol{\epsilon} = \frac{\frac{\mathbf{x}_{s_{m-1}}}{\alpha_{s_{m-1}}} - \frac{\mathbf{x}_{s_m}}{\alpha_{s_m}}}{\frac{\sigma_{s_{m-1}}}{\alpha_{s_{m-1}}} - \frac{\sigma_{s_m}}{\alpha_{s_m}}} = \frac{\Delta \mathbf{z}}{\Delta \text{NSR}}, \tag{6}$$

where $\Delta \mathbf{z}$ represents the change in $\mathbf{z}_t = \frac{\mathbf{x}_t}{\alpha_t}$, and $\Delta$NSR denotes the change in $\frac{\sigma_t}{\alpha_t}$. Once this noise $\boldsymbol{\epsilon}$ is calculated, it can be directly substituted into Equation 2 to compute $\mathbf{x}_t$ at any time $t$ along the ODE path.

At first glance, one might think that rectified diffusion (phased) is similar to the phased consistency model and related trajectory distillation methods . However, they differ in scope and application scenarios. From a technical perspective, although the learning objective for both is to compute the solution point for each segment, the construction of their inputs differs. For the phased consistency model, the sampled points still lie on the original PF-ODE trajectory, whereas in rectified diffusion (phased), the sampled points are constructed based on the properties of the first-order approximate ODE path as discussed above. In terms of application scenarios, the phased consistency model is limited by the constraints of the consistency model itself; to achieve deterministic multi-step inference, it typically requires training a separate model for each multi-step scenario. In contrast, rectified diffusion (phased) can theoretically enable deterministic inference for any number of steps (greater than a preset number of steps).

## 2.4 RECTIFIED DIFFUSION FACILITATES THE CONSISTENCY DISTILLATION

Previous work (Liu et al., 2023) proposes applying naive distillation after rectification to enhance one-step generation ability. This is because, after rectification, the model cannot achieve a perfect first-order path due to issues like optimization, model capacity, and ODE path intersections. As a result, rectified flow-based methods still do not perform as well as the most advanced distillation

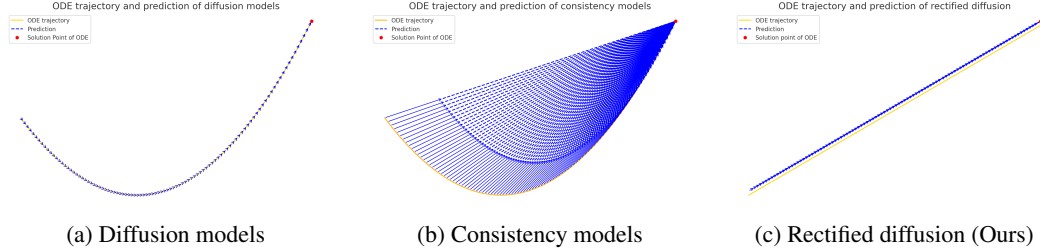

(a) Diffusion models  (b) Consistency models  (c) Rectified diffusion (Ours)

Figure 5: ODE trajectory and prediction comparison of consistency models and reftified diffusion. Since it's hard to visually tell whether a curved ODE path satisfies first-order property,*we apply straight lines for more clear demonstration.* The yellow line shows the ODE trajectories, while the blue line shows the predictions.

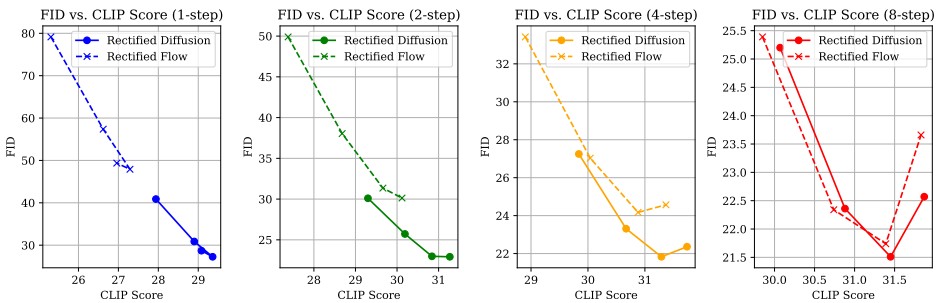

Figure 6: Effectiveness of Classifier-Free Guidance. The CFG values are 1, 1.2, 1.5, 2.0 respectively. By default, we adopt CFG value 1.5 for both rectified diffusion and rectified flow. Proper CFG values can significantly improve the performance even in one-step generation.

methods at low-step regime (e.g., 1-step generation). Following it, we also utilize distillation to further improve the model's performance at low-step regime after rectified diffusion. Instead of using naive distillation, we employ the more advanced distillation technique–consistency distillation (Song et al., 2023), which eliminates the need to regenerate large numbers of samples. Moreover, we found that after rectification, where the ODE path is close to the first-order approximate ODE path, consistency distillation leads to significantly faster training and better performance. This is because the training objective of a first-order approximate ODE path imposes a stronger constraint than self-consistency. In Fig. 5, we illustrate the differences between the diffusion model, consistency model, and rectified diffusion. The consistency model only adjusts the direction of the model's predictions without altering the ODE path itself, while rectified diffusion enforces a change in the ODE path.

## 3 EMPIRICAL VALIDATION

### 3.1 VALIDATION SETUP

To thoroughly compare our approach with rectified flow-based methods, we organize the comparison into three levels:

1) **Rectification comparison**: InstaFlow (Liu et al., 2023) proposes initializing a v-prediction-based flow-matching model using Stable Diffusion v1-5 (Rombach et al., 2022a), followed by further training with their rectified flow method, which we refer to as Rectified Flow. To compare with this, we apply the rectified diffusion method to continue training Stable Diffusion v1-5, referred to as Rectified Diffusion. This comparison aims to demonstrate the faster training speed and superior performance of our proposed rectified diffusion approach.

2) **Distillation comparison**: In the InstaFlow paper, the authors suggest using a standard distillation technique to improve the model's performance in a one-step scenario, which we refer to as Rectified Flow (Distill). Similarly, we apply a distillation strategy to enhance performance

at low-step regimes, specifically using consistency distillation to boost training efficiency. This approach is termed Rectified Diffusion (CD).

3) **Phased ODE segmentation**: PeRFlow (Yan et al., 2024) introduces the concept of segmenting the ODE and presents experimental results on both SD and SDXL (Podell et al., 2023), termed PeRFlow and PeRFlow-XL, respectively. We extend this idea by proposing a method for phasing the ODE to enforce first-order property within each sub-phase, which we call Rectified Diffusion (Phased) and Rectified Diffusion-XL (Phased).

Across all three of these comparative experiments, our methods demonstrate significantly superior performance.

Table 1: Performance comparison on validation set of COCO-2017.

| Method | Res. | Time ($\downarrow$) | # Steps | # Param. | FID ($\downarrow$) | CLIP ($\uparrow$) |
|---|---|---|---|---|---|---|
| SDv1-5+DPMSolver (Upper-Bound) (Lu et al., 2022) | 512 | 0.88s | 25 | 0.9B | 20.1 | 0.318 |
| Rectified Flow (Liu et al., 2023) | 512 | 0.88s | 25 | 0.9B | 21.65 | 0.315 |
| Rectified Flow (Liu et al., 2023) | 512 | 0.09s | 1 | 0.9B | 47.91 | 0.272 |
| Rectified Flow (Liu et al., 2023) | 512 | 0.13s | 2 | 0.9B | 31.35 | 0.296 |
| Rectified Diffusion (Ours) | 512 | 0.88s | 25 | 0.9B | 21.28 | 0.316 |
| Rectified Diffusion (Ours) | 512 | 0.09s | 1 | 0.9B | 27.26 | 0.295 |
| Rectified Diffusion (Ours) | 512 | 0.13s | 2 | 0.9B | 22.98 | 0.309 |
| Rectified Flow (Distill) (Liu et al., 2023) | 512 | 0.09s | 1 | 0.9B | 23.72 | 0.302 |
| Rectified Flow (Distill) (Liu et al., 2023) | 512 | 0.13s | 2 | 0.9B | 73.49 | 0.261 |
| Rectified Flow (Distill) (Liu et al., 2023) | 512 | 0.21s | 4 | 0.9B | 103.48 | 0.245 |
| Rectified Diffusion (CD) (Ours) | 512 | 0.09s | 1 | 0.9B | 22.83 | 0.305 |
| Rectified Diffusion (CD) (Ours) | 512 | 0.13s | 2 | 0.9B | 21.66 | 0.312 |
| Rectified Diffusion (CD) (Ours) | 512 | 0.21s | 4 | 0.9B | 21.43 | 0.314 |
| PeRFlow (Yan et al., 2024) | 512 | 0.21s | 4 | 0.9B | 22.97 | 0.294 |
| Rectified Diffusion (Phased) (Ours) | 512 | 0.21s | 4 | 0.9B | 20.64 | 0.311 |
| PeRFlow-SDXL (Yan et al., 2024) | 1024 | 0.71s | 4 | 3B | 27.06 | 0.335 |
| Rectified Diffusion-SDXL (Phased) (Ours) | 1024 | 0.71s | 4 | 3B | 25.81 | 0.341 |

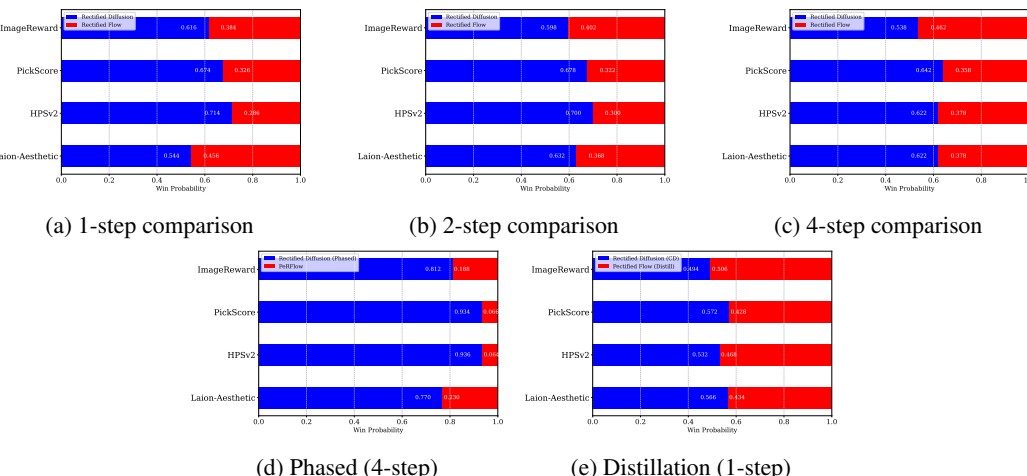

(a) 1-step comparison     (b) 2-step comparison     (c) 4-step comparison

(d) Phased (4-step)     (e) Distillation (1-step)

Figure 7: Human preference metrics comparison.

## 3.2 COMPARISON

**Training cost.** Following the setup from the InstaFlow paper, we first use Stable Diffusion v1-5 and DPM-Solver (Lu et al., 2022) to generate 1.6 million images. Since InstaFlow does not specify the prompts used, we generate images using a randomly sampled set of 1.6 million prompts. During the training of Rectified Diffusion, we used a batch size of 128 for a total of 200,000 iterations, resulting in a total of $128 \times 200,000 = 25,600,000$ samples processed. In comparison, InstaFlow processed $64 \times 70,000 + 1024 \times 25,000 = 30,080,000$ samples. Thus, our total training cost is lower than that

of InstaFlow. Additionally, InstaFlow's total training time was 75.2 A100 GPU days, whereas our method required approximately 20 A800 GPU days. Typically, the training efficiency of an A800 is about 80% of that of an A100. We attribute this significant reduction in training time to not using the LPIPS Loss (Zhang et al., 2018), which generally improves FID but incurs substantial memory and computational costs during the latent diffusion decoding process. *For the second-stage distillation,* we employ consistency distillation training with a batch size of 512 for 10,000 iterations, consuming a total of 4.6 A800 GPU days. In contrast, the distillation process described in the InstaFlow paper takes 110 A100 GPU days. Our training cost is approximately 3% of the GPU days of InstaFlow's distillation process.

**Training speed.** We monitor the performance of Rectified Diffusion in terms of FID and CLIP score at different stages of training. It was observed from Fig. 2 that our method achieve superior one-step performance compared to Rectified Flow after just 20,000 iterations, with further significant improvements as training continued. At this stage, the number of samples processed was only about 8% of the samples processed by Rectified Flow. This efficiency is largely because Rectified Diffusion does not require converting the original epsilon prediction diffusion model, which follows the DDPM form, into a $v$-prediction flow-matching model—a process that incurs significant computational cost.

**Qualitative comparison.** We present a comparison of the images generated by Rectified Diffusion and Rectified Flow across various scenarios in Fig. 8 and Fig. 9. First, we can observe that the Rectified Flow model performs poorly at low step counts, producing only very blurry images in fewer than eight steps. Additionally, we notice that the images generated by PeRFlow are blurry and fail to reflect the content of the text. Moreover, the results generated by Rectified Flow (Distill) remain relatively blurry and lack the ability for multi-step refinement, which limits its applicability. Rectified Diffusion shows clearly superiority in these settings.

**Quantitative comparison.** We calculate the FID (Heusel et al., 2017) and CLIP scores (Radford et al., 2021) for different models on the COCO-2017 validation set (Lin et al., 2014) and the 30k subset of the COCO-2014 validation set (Lin et al., 2014), respectively. As shown in Table 1 and Table 3, our model consistently outperforms the methods based on rectified flow across both metrics, different scenarios, and various steps. It also achieves performance comparable to advanced distillation and GAN training methods.

**Human preference metrics.** To more comprehensively evaluate the model performance, we compare the outputs using human preference models. We follow the testing setup of Diffusion-DPO (Wallace et al., 2024), generating images with 500 unique prompts from the Pick-a-pic (Kirstain et al., 2023) validation set for comparison. We used the Laion-Aesthetic Predictor (Schuhmann, 2022), Pickscore (Kirstain et al., 2023), HPSv2 (Wu et al., 2023), and ImageReward (Xu et al., 2024a) to score the generated results from each model individually and calculate the win rate of each model across these metrics. Our results, shown in Fig 7, consistently outperform the results of rectified flow-based models.

**CFG-influence.** We show the performance comparison of FID and CLIP Score between Rectified Flow and Rectified Diffusion under different step counts and CFG values in Fig. 6. We observe that Rectified Diffusion consistently outperforms Rectified Flow, especially in the low-step regime. Additionally, we find that CFG has a significant impact on both Rectified Diffusion and Rectified Flow; even in the 1-step generation scenario, using an appropriate CFG value can still significantly enhance performance. The CFG values are 1, 1.2, 1.5, 2.0 respectively.

## 4 CONCLUSION

In conclusion, we rethink and investigate the essence of rectified flow. We demonstrate that retraining with pre-collected noise-image pairs is the most important factor. Building on this insight, we propose Rectified Diffusion, extending its scope to general diffusion forms. We identify that it is not straightness but first-order property is the essential training target of Rectified Diffusion. Additionally, by incorporating consistency distillation and introducing Rectified Diffusion (Phased), we further enhance training efficiency and model performance, offering a streamlined approach to efficient high-fidelity visual generation. Vast validation demonstrates the advancements of Rectified Diffusion.

ACKNOWLEDGEMENTS

This project is funded in part by National Key R&D Program of China Project 2022ZD0161100, by the Centre for Perceptual and Interactive Intelligence (CPII) Ltd under the Innovation and Technology Commission (ITC)'s InnoHK, by NSFC-RGC Project N_CUHK498/24. Hongsheng Li is a PI of CPII under the InnoHK.

REPRODUCIBILITY STATEMENT

We have undertaken substantial efforts to ensure that the results in this paper are reproducible. The training and evaluation code, along with detailed guidance, is made available at `https://github.com/G-U-N/Rectified-Diffusion`. Additionally, we have released the pre-trained weights at `https://huggingface.co/wangfuyun/Rectified-Diffusion`. We believe these resources will aid in replicating our findings and foster further research that builds upon our contributions.

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

# APPENDIX

## I RELATED WORKS

**Diffusion models.** Diffusion models have steadily become the foundational models in image synthesis (Ho et al., 2020; Song et al., 2020b; Karras et al., 2022). Extensive research has been conducted to explore their underlying principles (Lipman et al., 2022; Chen & Lipman, 2023; Song et al., 2020b; Kingma et al., 2021; Chen et al., 2023; 2024; Guo et al., 2025; Yuan et al., 2023) and to expand or enhance the design space of these models (Song et al., 2020a; Karras et al., 2022; Kingma et al., 2021). Additionally, several works have focused on innovating the model architecture (Dhariwal & Nichol, 2021; Peebles & Xie, 2023), while others have scaled up diffusion models for text-conditioned image synthesis and various real-world applications (Shi et al., 2024; Rombach et al., 2022b; Podell et al., 2023; Wang et al., 2024e;c;f;d). Moreover, efforts to accelerate sampling have been pursued at both the scheduler level (Karras et al., 2022; Lu et al., 2022; Song et al., 2020a) and the training level (Meng et al., 2023; Song et al., 2023; Zhou et al., 2024b;a). The former typically involves refining the approximation of the PF-ODE (Lu et al., 2022; Song et al., 2020a), while the latter focuses on distillation techniques (Meng et al., 2023; Salimans & Ho, 2022; Song et al., 2023; Wang et al., 2024b;c;a; Mao et al., 2024; Geng et al., 2024) or initializing diffusion weights for GAN training (Sauer et al., 2023c; Lin et al., 2024; Xu et al., 2024b).

**Rectified flow.** Lipman et al. (2022) proposes the flow matching based on continuous normalizing flows, providing a different and unified perspective to understand diffusion models. Liu et al. (2022) proposes the method rectified flow, setting up important baseline for diffusion acceleration and providing a solid theoretical analysis. It proposes rectification to straighten the ODE path of flow-matching based diffusion models. In the proof, Liu et al. (2022) show that the rectification allows for non-decreasing straightness of ODE. Liu et al. (2023) scale up the idea of rectified flow into large text-to-image generations, achieving one-step generation without introducing GAN. Yan et al. (2024) and Yang et al. (2024) propose to split the ODE path into multi-phase following the InstaFlow (Liu et al., 2023). Lee et al. (2024) analysises that one-time rectification is generally enough to achieve pure straightness and proposes better optimization strategy for enhanced performance of rectified flows.

**Stochastic interpolants.** Stochastic Interpolants (Albergo et al., 2023; Albergo & Vanden-Eijnden, 2022) unifies flow-based and diffusion-based methods by extending the framework of continuous-time normalizing flow with continuous-time stochastic processes that bridge arbitrary probability densities in finite time, enabling flexible, noise-adjustable deterministic and stochastic generation while linking to score-based diffusion and Schrodinger bridges.

**Noise coupling.** In addition to rectified flow, which constructs pairwise relationships between data and noise using pretrained diffusion models, there are several other methods that reduce the variance of noise-sample pairings by measuring the distance between noise and samples. For instance, OT-CFM (Tong et al., 2023) and Multisample Flow Matching (Pooladian et al., 2023) propose building a pairing probability matrix by computing the optimal transport relationship between samples and noise within a batch. Similarly, Fast-ODE (Lee et al., 2023) suggests training a variational encoder to model a data-dependent noise distribution during training. Although these techniques have been validated within the context of flow models, our analysis in this paper suggests that they are broadly applicable to the general framework of diffusion models. Additionally, SFT (Xu et al., 2023) and SCT (Xu et al., 2023) numerically solve the variance-reduced score function of the data distribution, significantly lowering the training variance in both diffusion models and consistency models.

## II PROOF FOR FIRST-ORDER APPROXIMATE ODE PATH

**Theorem 1** *For the general diffusion form* $\mathbf{x}_t = \alpha_t \mathbf{x}_0 + \sigma_t \boldsymbol{\epsilon}$, *there exists an exact ODE solution form as follows:*

$$\mathbf{x}_t = \frac{\alpha_t}{\alpha_s}\mathbf{x}_s - \alpha_t \int_{\lambda_s}^{\lambda_t} e^{-\lambda} \boldsymbol{\epsilon}_{\boldsymbol{\theta}}(\mathbf{x}_{t_\lambda}, t_\lambda)\mathrm{d}\lambda, \tag{7}$$

*where* $\lambda_t = \ln\frac{\alpha_t}{\sigma_t}$ *and* $t_\lambda$ *is the inverse function of* $\lambda_t$. *The first-order approximate ODE path satisfies*

$$\mathbf{x}_t = \frac{\alpha_t}{\alpha_s}\mathbf{x}_s - \alpha_t \boldsymbol{\epsilon}_{\boldsymbol{\theta}}(\mathbf{x}_s, s) \int_{\lambda_s}^{\lambda_t} e^{-\lambda}\mathrm{d}\lambda = \frac{\alpha_t}{\alpha_s}\mathbf{x}_s - \alpha_t \boldsymbol{\epsilon}_{\boldsymbol{\theta}}(\mathbf{x}_s, s)\Big(\frac{\alpha_s}{\sigma_s} - \frac{\alpha_t}{\sigma_t}\Big). \tag{8}$$

*We show the equivalence between Equation 7 and Equation 8 for arbitrary* $t$ *and* $s$, *which holds true* *if and only if* $\boldsymbol{\epsilon}_{\boldsymbol{\theta}}(\mathbf{x}_t, t)$ *is constant.*

**Proof 1** *If* $\boldsymbol{\epsilon}_{\boldsymbol{\theta}}(\mathbf{x}_t, t)$ **is constant, then the Equation 7 and Equation 8 are equivalent.**

*Assumption: Let* $\boldsymbol{\epsilon}_{\boldsymbol{\theta}}(\mathbf{x}_s, s) = \boldsymbol{\epsilon}_0$ *be a constant.*

*Substituting* $\boldsymbol{\epsilon}_0$ *into the Equation 7:*

$$\mathbf{x}_t = \frac{\alpha_t}{\alpha_s}\mathbf{x}_s - \alpha_t \boldsymbol{\epsilon}_0 \int_{\lambda_s}^{\lambda_t} e^{-\lambda}\mathrm{d}\lambda \tag{9}$$

*Calculating the integral:*

$$\int_{\lambda_s}^{\lambda_t} e^{-\lambda}\mathrm{d}\lambda = e^{-\lambda_s} - e^{-\lambda_t} = \frac{\sigma_s}{\alpha_s} - \frac{\sigma_t}{\alpha_t} \tag{10}$$

*Substituting the result:*

$$\mathbf{x}_t = \frac{\alpha_t}{\alpha_s}\mathbf{x}_s - \alpha_t \boldsymbol{\epsilon}_0 \left(\frac{\sigma_s}{\alpha_s} - \frac{\sigma_t}{\alpha_t}\right) \tag{11}$$

*Comparing with the equation: The results match, thus proving equivalence.*

**If Equation 7 and Equation 8 are equivalent, then** $\boldsymbol{\epsilon}_{\boldsymbol{\theta}}(\mathbf{x}_t, t)$ **must be constant.**

*Assumption: Assume the two are equivalent:*

$$-\alpha_t \int_{\lambda_s}^{\lambda_t} e^{-\lambda} \boldsymbol{\epsilon}_{\boldsymbol{\theta}}(\mathbf{x}_{t_\lambda}, t_\lambda)\mathrm{d}\lambda = -\alpha_t \boldsymbol{\epsilon}_{\boldsymbol{\theta}}(\mathbf{x}_s, s) \int_{\lambda_s}^{\lambda_t} e^{-\lambda}\mathrm{d}\lambda \tag{12}$$

*Removing the constant factor:*

$$\int_{\lambda_s}^{\lambda_t} e^{-\lambda} \boldsymbol{\epsilon_\theta}(\mathbf{x}_{t_\lambda}, t_\lambda) \mathrm{d}\lambda = \boldsymbol{\epsilon_\theta}(\mathbf{x}_s, s) \int_{\lambda_s}^{\lambda_t} e^{-\lambda} \mathrm{d}\lambda \tag{13}$$

*Differentiating with respect to t with Newton-Leibniz theorem:*

$$\frac{d}{dt}\left(\int_{\lambda_s}^{\lambda_t} e^{-\lambda} \boldsymbol{\epsilon_\theta}(\mathbf{x}_{t_\lambda}, t_\lambda) \mathrm{d}\lambda\right) = e^{-\lambda_t} \boldsymbol{\epsilon_\theta}(\mathbf{x}_{t_\lambda}, t_\lambda) \frac{d\lambda_t}{dt} \tag{14}$$

*Comparing both sides:*

$$e^{-\lambda_t} \boldsymbol{\epsilon_\theta}(\mathbf{x}_{t_\lambda}, t_\lambda) \frac{d\lambda_t}{dt} = \boldsymbol{\epsilon_\theta}(\mathbf{x}_s, s) e^{-\lambda_t} \frac{d\lambda_t}{dt} \tag{15}$$

*As indicated by previous work (Kingma et al., 2021), $\mathbf{SNR}(t) = \frac{\alpha_t^2}{\sigma_t^2}$ is the monotonically decreasing function. Therefore, we have $\lambda_t = \ln\frac{\alpha_t}{\sigma_t} = \frac{1}{2}\ln\mathbf{SNR}(t)$ is a monotonically decreasing function. Therefore $\frac{\mathrm{d}\lambda_t}{\mathrm{d}_t} < 0$. Since $\frac{d\lambda_t}{dt} \neq 0$ and $e^{-\lambda_t} > 0$, we can cancel terms, leading to:*

$$\boldsymbol{\epsilon_\theta}(\mathbf{x}_{t_\lambda}, t_\lambda) = \boldsymbol{\epsilon_\theta}(\mathbf{x}_s, s), \forall t_\lambda \in [s, t]. \tag{16}$$

*This shows that for any $t$, $\boldsymbol{\epsilon_\theta}(\mathbf{x}_t, t)$ must be constant, proving the "if and only if" statement.*

## III   LIMITATION

At low-step regime, the performance of methods based on rectification still lags behind state-of-the-art methods based on distillation (Zhou et al., 2024b) or GAN training (Yin et al., 2024b; Sauer et al., 2023c). Additional distillation steps are needed to improve low-step performance, which is also stated in InstaFlow (Liu et al., 2023).

## IV   WHY PERFECT COUPLING LEADS TO FIRST-ORDER APPROXIMATE ODE PATH?

For a diffusion model trained with the form $\mathbf{x}_t = \alpha_t \mathbf{x}_0 + \sigma_t \boldsymbol{\epsilon}$ (Kingma et al., 2021). The score $\boldsymbol{s_\theta}$ will converge to the expectation value of all the possible conditional scores $\nabla_{\mathbf{x}_t} \log \mathbb{P}(x_t|\mathbf{x}_0)$ which is determined by data $\mathbf{x}_0$ and noise $\boldsymbol{\epsilon}$ ($\nabla_{\mathbf{x}_t} \log \mathbb{P}(x_t|\mathbf{x}_0) = -\frac{\mathbf{x}_t - \alpha_t \mathbf{x}_0}{\sigma_t^2}$) due to the noise $\boldsymbol{\epsilon}$ and data $\mathbf{x}_0$ is randomly paired (Song et al., 2020b). That is

$$\boldsymbol{s_\theta}(\mathbf{x}_t, t) = \mathbb{E}_{\mathbb{P}(\mathbf{x}_0|\mathbf{x}_t)}[\nabla_{\mathbf{x}_t} \log \mathbb{P}(\mathbf{x}_t|\mathbf{x}_0)]. \tag{17}$$

The expectation score value on the same PF-ODE but different timesteps will converge to different values and directions (Song et al., 2020b). Therefore, on the PF-ODE, the score $\boldsymbol{s_\theta}(\mathbf{x}_t, t)$ will change along the time axis. Since $\boldsymbol{\epsilon} = -\sigma_t \boldsymbol{s_\theta}$, it means that the epsilon prediction will change along the time axis, which is not comprised with our theorem 1. It is not a first-order approximate ODE path.

However, if we achieve perfect noise data coupling and satisfy the no-intersection condition, $\mathbf{x}_t$ will be sampled with a single noise and data pair. Without optimization errors, we will have

$$\boldsymbol{s_\theta}(\mathbf{x}_t, t) = \nabla_{\mathbf{x}_t} \log \mathbb{P}(\mathbf{x}_t \mid \mathbf{x}_0) = -\frac{\mathbf{x}_t - \alpha_t \mathbf{x}_0}{\sigma_t^2} = -\frac{\boldsymbol{\epsilon}}{\sigma_t}, \tag{18}$$

Since $\boldsymbol{\epsilon_\theta} = -\sigma_t \boldsymbol{s_\theta}$, therefore the epsilon prediction on the PF-ODE should always be the $\boldsymbol{\epsilon}$. Return to our Theorem 1, that is to say, we achieve the first-order approximate ODE path.

## V   VALIDATION

### V.1   EMPIRICAL EVIDENCE FOR THE SUPERIORITY OF RECTIFIED DIFFUSION.

**Rectified Diffusion outperforms Rectified Flow with significantly smaller training costs.** As highlighted in Fig. 2, where we show the performance change as the training iterations. Our one-step performance significantly outperforms the one-step performance of the official weight of Rectified Flow (Liu et al., 2023) within only 20, 000 iterations with batch size 128. The overall number of trained images at this iteration is only 8% of the number of trained images in Rectified Flow. Even after the full training of 200, 000 iterations, considering that our batch size adopted is only 128 which is significantly smaller than the batch size adopted in previous work, we still use less trained images than the official weight of Rectified Flow method.

**Rectified Diffusion significantly outperforms Rectified Flow in the few-step setting.** The performance in the few-step setting is the most important criterion to judge the effectiveness of rectification on the ODE (since we aim to achieve efficient sampling and improve the performance in few-step sampling). Rectified Diffusion significantly outperforms Rectified Flow in this setting. Specifically, on COCO-5K. Rectified Flow achieves one-step FID 47.91, and two-step FID 31.35. Instead, Rectified Diffusion achieves one-step FID 27.26 (significantly outperforms the two-step FID of Rectified Flow), and two-step FID 22.98 (approaching the 25-step FID of Rectified Flow 21.65). The is strong evidence that Rectified Diffusion is better than Rectified Flow.

**Rectified Diffusion has better performance upper bound than Rectified Flow.** It should be noted that Rectified Diffusion and Rectified Flow are all trained with generated data from pretrained diffusion models. Specifically, in the default setting, they both use 25-step DPMSolver with Stable Diffusion v1-5 to generate noise-data pairs. Therefore, the performance of 25-step DPMSolver with the teacher diffusion model works as the performance upper-bound. The 25-step Rectified Diffusion achieves better performance than 25-step Rectified Flow on both FID and CLIP SCORE. Therefore, Rectified Diffusion has better performance upper bound than Rectified Flow. Additionally, please note that the 25-step performance of Rectified Diffusion and that of Rectified Flow are already very close to the 25-step DPMSolver with the teacher diffusion model. Therefore, we can not achieve a very large improvement.

**Rectified Diffusion (Phased) consistently surpasses the baseline PeRFlow (Yan et al., 2024).** We extend Rectified Diffusion into the phased setting, and our performance still consistently outperforms the previous phased rectified flow-based method.

**Rectified Diffusion (CD) surpasses the baseline Rectified Flow (Distill) with only 3% GPU days for training.** The distillation process described in the InstaFlow (Liu et al., 2023) takes 110 A100 GPU days. Our training cost is approximately 3% of the GPU days of InstaFlow's distillation process. Moreover, the results generated by Rectified Flow (Distill) remain relatively blurry and lack the ability for multi-step refinement, which limits its applicability. Instead, Rectified Diffusion (CD) not only achieves better one-step performance but also supports multistep refinement to further improve the performance.

### V.2   MORE VALIDATION ON ADDITIONAL DATASETS.

We additionally evaluate model performance on the Laion (Schuhmann, 2022) and CC3M (Changpinyo et al., 2021) subsets. Following the test setting of COCO-2017, we adopt the 5k subset for evaluation. Our experimental results as shown in Table 2 show that Rectified Diffusion consistently outperforms Rectified Flow, consistent with our original evaluation on the COCO dataset. This provides strong evidence of the superiority of Rectified Diffusion.

## VI   MORE DISCUSSION

### VI.1   MOTIVATION AND CONTRIBUTIONS

The motivation for rectified diffusion is to investigate and rethink the most essential part of rectified flow (Liu et al., 2022). The investigation allows us to extend the scope of rectification, which was

Table 2: Performance comparison on Laion and CC3M subsets.

| Subset | Step | Metric | Rectified Diffusion | Rectified Flow |
|--------|------|--------|---------------------|----------------|
| Laion | 8 | CLIP Score | 26.36 | 25.49 |
| | | FID | 23.52 | 24.73 |
| | 4 | CLIP Score | 25.80 | 24.65 |
| | | FID | 24.70 | 27.27 |
| | 2 | CLIP Score | 25.15 | 23.38 |
| | | FID | 26.14 | 35.09 |
| | 1 | CLIP Score | 23.30 | 20.12 |
| | | FID | 30.14 | 52.86 |
| CC3M | 8 | CLIP Score | 28.33 | 27.65 |
| | | FID | 28.10 | 29.07 |
| | 4 | CLIP Score | 28.08 | 27.37 |
| | | FID | 29.97 | 31.54 |
| | 2 | CLIP Score | 27.68 | 26.27 |
| | | FID | 31.56 | 37.72 |
| | 1 | CLIP Score | 26.30 | 24.63 |
| | | FID | 34.28 | 49.06 |

originally proposed and believed only suitable for rectified flow (using the form $\mathbf{x}_t = (1-t)\mathbf{x}_0 + t\boldsymbol{\epsilon}$, to general diffusion models.

In particular, InstaFlow (Liu et al., 2023), an important follow-up work to rectified flow that extends it to text-to-image tasks and serves as a key baseline in our paper, uses the Stable Diffusion (Which is an $\epsilon$-prediction neural network follows DDPM diffusion form) for initialization. However, in InstaFlow, the model is first converted into a $v$-prediction flow-matching model before undergoing retraining of paired noise-sample This process introduces a gap between the pretrained model and the retrained model supporting efficient sampling.

The conversion that converts the pretrained diffusion models (can be forms of DDPM, Sub-VP, VE (Song et al., 2020b)) has become a common practice in the important following works (Liu et al., 2022; Yan et al., 2024; Lee et al., 2024).

**To our knowledge, we are the first to point out that this conversion is unimportant and show that straightness is not the essential training target. Any ODE (even curved) can achieve one-step sampling as long as it satisfies the requirement of first-order approximate property.**

Therefore, our main contribution is to rethink and break the incomplete understanding from previous research. Additionally, we make our method as simple, straightforward, and accessible as possible to most diffusion-related researchers. Essentially, rectified diffusion does not need to change anything of pretrained diffusion models for sampling acceleration (including networks, preconds, training/inference code, noise scheduler, etc.). The only difference as highlighted in our paper is to replace the noise and data in standard diffusion training with paired noise (collected) and data (generated).

With this simple design, we still achieve significant improvement compared to the previous best-performing rectified flow-based methods (with even smaller training costs) and comparable performance to previous best-performing distillation or GAN-based acceleration methods. We believe this discovery can inspire related research and broaden the understanding of diffusion models in the community.

### VI.2 CLARIFICATION ON THE DIFFERENCE BETWEEN RECTIFIED DIFFUSION AND CONSISTENCY DISTILLATION.

The crucial difference between rectification (aims to achieve first-order approximate ODE path) and distillation (aims to directly predict the solution point of PF-ODE ) is that rectification will change the ODE trajectory but distillation does not. We provided a visualization explanation in Fig. 5. Since it's hard to intuitively determine whether a curved ODE path satisfies first-order property, we represent the first-order approximate ODE path with a straight line (It is not straight in most cases). Consistency models change the prediction of original diffusion models but do not change the ODE trajectory. This makes it only suitable for stochastic multistep sampling (If it uses diffusion

samplers like DDIM (Song et al., 2020a), it will fall out from the ODE trajectory, causing the further prediction to be degraded). Instead, rectification smoothens and rectifies the ODE trajectory to make it the first-order approximate ODE path. Generally, after rectification, the diffusion model is still a diffusion model (since its prediction direction still follows the derivative of its ODE as shown in Fig. 5), but performs much better in the few-step setting. Instead, diffusion models after consistency distillation will become consistency models, since its prediction direction is different from the derivative of its PF-ODE.

### VI.3 CLARIFICATION ON THE ADDITIONAL DISTILLATION PROCEDURE AFTER RECTIFICATION.

**To set up a fair comparison.** The most direct reason why we adopted distillation to further improve performance is to set up a fair comparison with InstaFlow (Liu et al., 2023). The difference is that we adopt consistency distillation instead of the naive distillation strategy adopted in InstaFlow. We achieve better 1-step performance than the distilled baseline in InstaFlow with only 3% GPU days used by InstaFlow for distillation.

**Rectification is a harder objective than distillation.** The root reason is that first-order approximate ODE path is a harder target than distillation. Intuitively speaking, since the first-order approximate ODE path is a stronger constraint than self-consistency, it is harder to train and will face more optimization issues. However, despite the difficulty of achieving perfect first-order approximate ODE paths, after rectification, the ODE of our model will become generally closer to the first-order approximate ODE path. It will reduce the difficulty of further distillation if we use the model that has undergone rectification as the teacher model for distillation.

### VI.4 CLARITY ON THE DIFFERENCE BETWEEN DPM-SOLVER AND RECTIFIED DIFFUSION

Following the proof of DPM-Solver-1 (Equation 3.6 and the subsequent equation on Page 5 of DPM-Solver), which is equivalent to the DDIM sampler, we can approximate the ODE solution from time $t$ to $s$ as follows:

$$\mathbf{x}_s = \frac{\alpha_s}{\alpha_t}\mathbf{x}_t - \alpha_s\boldsymbol{\epsilon}(\mathbf{x}_t, t)\int_{\lambda_t}^{\lambda_s} e^{-\lambda}\mathrm{d}\lambda + \mathcal{O}((\lambda_s - \lambda_t)^2). \tag{19}$$

In DPM-Solver-1, higher-order error terms $\mathcal{O}((\lambda_s - \lambda_t)^2)$ are neglected during inference, which becomes problematic when $\lambda_s - \lambda_t$ is large. **This is the primary reason why DPM-Solver-1 (DDIM) requires at least 50 steps for high-quality generation.**

In contrast, our paper does not rely on such level approximations. Our primary goal is to rectify the PF-ODE to ensure it satisfies first-order accuracy, **which eliminates the high-order error terms** through paired retraining (Rectification). This is the fundamental reason why **Rectified Diffusion can achieve 1-step generation**. In comparison, DPM-Solver requires at least several steps—typically over ten—for high-quality generation.

## VII MORE RESULTS

Table 3: Performance comparison on COCO-2014.

| Method | Res. | Time (↓) | # Steps | # Param. | FID (↓) | CLIP (↑) |
|---|---|---|---|---|---|---|
| Autoregressive Models | | | | | | |
| DALL·E (Ramesh et al., 2021) | 256 | - | - | 12B | 27.5 | - |
| CogView2 (Ding et al., 2021) | 256 | - | - | 6B | 24.0 | - |
| Parti-750M (Yu et al., 2022) | 256 | - | - | 750M | 10.71 | - |
| Parti-3B (Yu et al., 2022) | 256 | 6.4s | - | 3B | 8.10 | - |
| Parti-20B (Yu et al., 2022) | 256 | - | - | 20B | 7.23 | - |
| Make-A-Scene (Gafni et al., 2022) | 256 | 25.0s | - | - | 11.84 | - |
| Masked Models | | | | | | |
| Muse (Chang et al., 2023) | 256 | 1.3 | 24 | 3B | 7.88 | 0.32 |
| Diffusion Models | | | | | | |
| GLIDE (Nichol et al., 2021) | 256 | 15.0s | 250 | 5B | 12.24 | - |
| DALL·E 2 (Ramesh et al., 2022) | 256 | - | 250+27 | 5.5B | 10.39 | - |
| LDM (Rombach et al., 2022a) | 256 | 3.7s | 250 | 1.45B | 12.63 | - |
| Imagen (Saharia et al., 2022) | 256 | 9.1s | - | 3B | 7.27 | - |
| eDiff-I (Balaji et al., 2022) | 256 | 32.0s | 25+10 | 9B | 6.95 | - |
| Generative Adversarial Networks (GANs) | | | | | | |
| LAFITE (Zhou et al., 2022) | 256 | 0.02s | 1 | 75M | 26.94 | - |
| StyleGAN-T (Sauer et al., 2023a) | 512 | 0.10s | 1 | 1B | 13.90 | ∼0.293 |
| GigaGAN (Kang et al., 2023) | 512 | 0.13s | 1 | 1B | 9.09 | - |
| Stable Diffusion (0.9 B) and its accelerated or distilled versions | | | | | | |
| GANs | | | | | | |
| UFOGen (Xu et al., 2024b) | 512 | 0.09s | 1 | 0.9B | 12.78 | - |
| DMD (CFG=3) (Yin et al., 2024a) | 512 | 0.09s | 1 | 0.9B | 11.49 | - |
| DMD (CFG=8) (Yin et al., 2024a) | 512 | 0.09s | 1 | 0.9B | 14.98 | 0.320 |
| SD-Turbo (Sauer et al., 2023c) | 512 | 0.09s | 1 | 0.9B | 16.59 | 0.312 |
| Distillation | | | | | | |
| BOOT (Gu et al., 2023) | 512 | 0.09s | 1 | 0.9B | 48.20 | 0.26 |
| Guided Distillation (Meng et al., 2023) | 512 | 0.09s | 1 | 0.9B | 37.3 | 0.27 |
| LCM (Luo et al., 2023) | 512 | 0.09s | 1 | 0.9B | 37.3 | 0.27 |
| Phased Consistency Model (Wang et al., 2024b) | 512 | 0.09s | 1 | 0.9B | 17.91 | 0.296 |
| Phased Consistency Model (Wang et al., 2024b) | 512 | 0.21s | 4 | 0.9B | 11.70 | - |
| SiD-LSG ($\kappa = 4.5$) | 512 | 0.09s | 1 | 0.9B | 16.59 | 0.317 |
| SiD-LSG ($\kappa = 3$) | 512 | 0.09s | 1 | 0.9B | 13.21 | 0.314 |
| SiD-LSG ($\kappa = 2$) | 512 | 0.09s | 1 | 0.9B | 9.56 | 0.313 |
| SiD-LSG ($\kappa = 1.5$) | 512 | 0.09s | 1 | 0.9B | 8.71 | 0.302 |
| SiD-LSG ($\kappa = 4.5$) | 512 | 0.09s | 1 | 0.9B | 16.59 | 0.317 |
| Rectification (⋆ ⋆ ⋆) | | | | | | |
| SDv1-5+DPMSolver (Upper-Bound) (Lu et al., 2022) | 512 | 0.88s | 25 | 0.9B | 9.78 | 0.318 |
| Rectified Flow (Liu et al., 2023) | 512 | 0.88s | 25 | 0.9B | 11.34 | 0.313 |
| Rectified Flow (Liu et al., 2023) | 512 | 0.09s | 1 | 0.9B | 36.68 | 0.272 |
| Rectified Flow (Liu et al., 2023) | 512 | 0.13s | 2 | 0.9B | 20.01 | 0.296 |
| Rectified Diffusion (Ours) | 512 | 0.88s | 25 | 0.9B | 10.73 | 0.315 |
| Rectified Diffusion (Ours) | 512 | 0.09s | 1 | 0.9B | 16.88 | 0.293 |
| Rectified Diffusion (Ours) | 512 | 0.13s | 2 | 0.9B | 12.57 | 0.307 |
| Rectified Flow (Distill) (Liu et al., 2023) | 512 | 0.09s | 1 | 0.9B | 13.67 | 0.302 |
| Rectified Flow (Distill) (Liu et al., 2023) | 512 | 0.13s | 2 | 0.9B | 62.81 | 0.261 |
| Rectified Diffusion (CD) (Ours) | 512 | 0.09s | 1 | 0.9B | 12.54 | 0.303 |
| Rectified Diffusion (CD) (Ours) | 512 | 0.13s | 2 | 0.9B | 11.41 | 0.310 |
| PeRFlow (Yan et al., 2024) | 512 | 0.21s | 4 | 0.9B | 18.59 | 0.264 |
| Rectified Diffusion (Phased) (Ours) | 512 | 0.21s | 4 | 0.9B | 10.21 | 0.310 |
| Stable Diffusion XL (3B) and its accelerated or distilled versions | | | | | | |
| GANs | | | | | | |
| SDXL-Turbo Sauer et al. (2023c) | 512 | 0.15s | 1 | 3B | 24.57 | 0.337 |
| SDXL-Turbo Sauer et al. (2023c) | 512 | 0.34s | 4 | 3B | 23.19 | 0.334 |
| SDXL-Lightning (Lin et al., 2024) | 1024 | 0.35s | 1 | 3B | 23.92 | 0.316 |
| SDXL-Lightning (Lin et al., 2024) | 1024 | 0.71s | 4 | 3B | 24.56 | 0.323 |
| DMDv2 (Yin et al., 2024b) | 1024 | 0.35s | 1 | 3B | 19.01 | 0.336 |
| DMDv2 (Yin et al., 2024b) | 1024 | 0.71s | 4 | 3B | 19.32 | 0.332 |
| Distillation | | | | | | |
| LCM (Luo et al., 2023) | 1024 | 0.35s | 1 | 3B | 81.62 | 0.275 |
| LCM (Luo et al., 2023) | 1024 | 0.71s | 4 | 3B | 22.16 | 0.317 |
| Phased Consistency Model (Wang et al., 2024b) | 1024 | 0.35s | 1 | 3B | 25.31 | 0.318 |
| Phased Consistency Model (Wang et al., 2024b) | 1024 | 0.71s | 4 | 3B | 21.04 | 0.329 |
| Rectification (⋆ ⋆ ⋆) | | | | | | |
| PeRFlow-XL (Yan et al., 2024) | 1024 | 0.71s | 4 | 3B | 20.99 | 0.334 |
| Rectified Diffusion-XL (Phased) (Ours) | 1024 | 0.71s | 4 | 3B | 19.71 | 0.340 |

Results of Stable Diffusion XL-based models are tested with COCO-2014 10k following the evaluation setting of DMDv2 (Yin et al., 2024b). Other results are tested with COCO-2014 30k following the karpathy split.

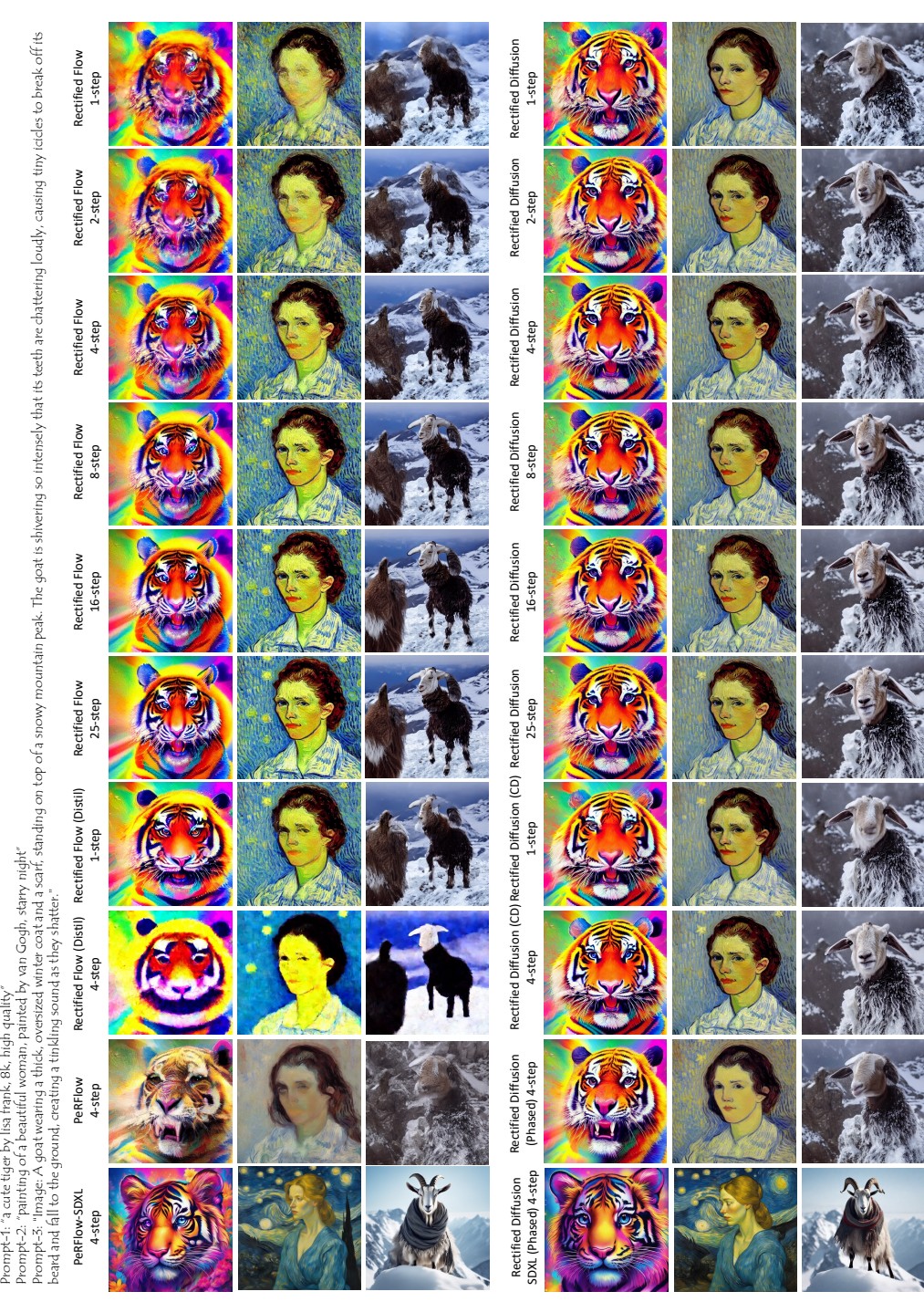

Figure 8: Qualitative comparison.

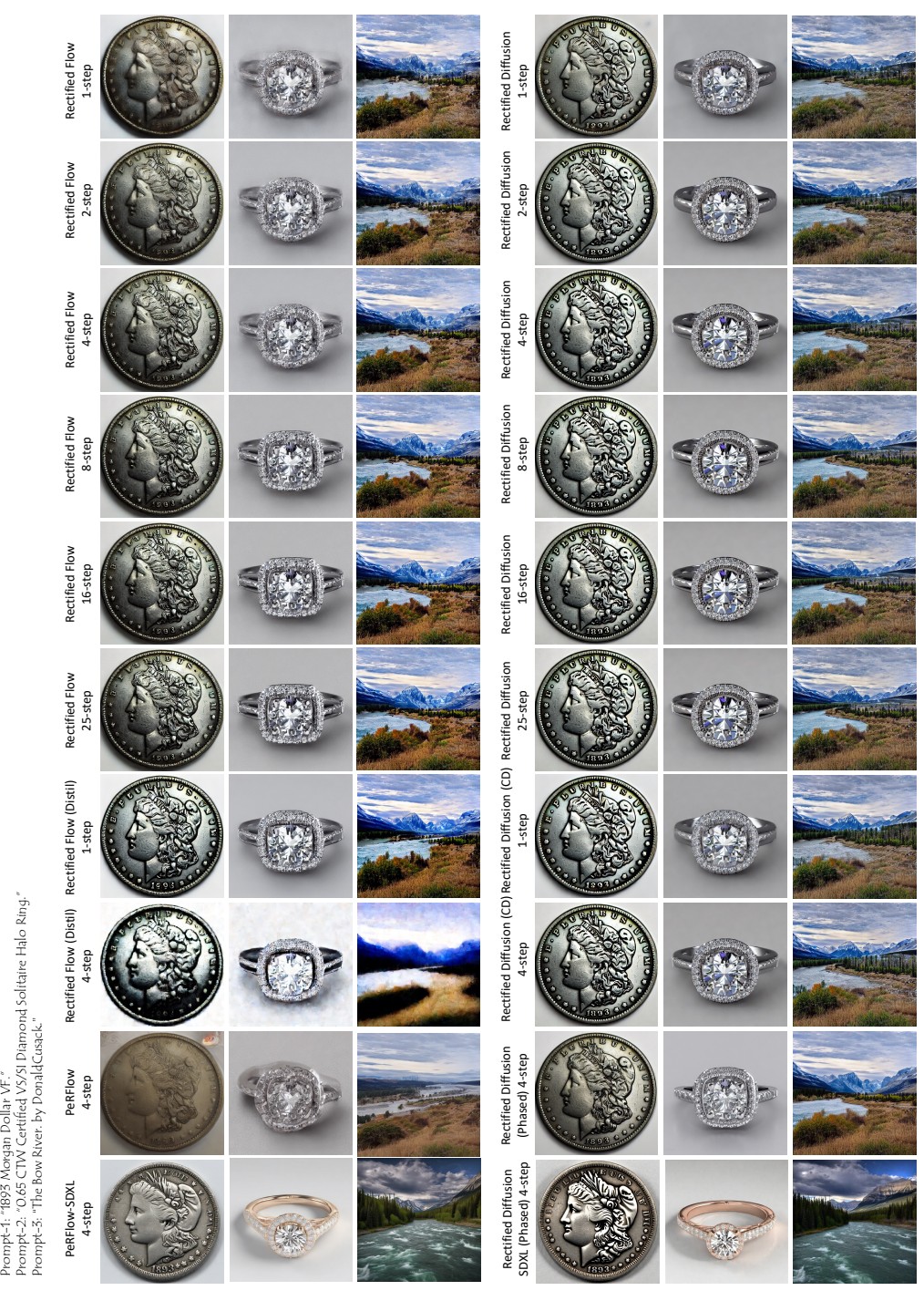

Figure 9: Qualitative comparison.

Prompt: "sill life photo of an apple."

Prompt: "A cat in a space suit walking on the moon."

Prompt: "guinea pigs on a pirate ship."

Prompt: "a husky running on the beach."

Prompt: "insanely detailed portrait,female model, insane face details, perfect eyes,dof, dslr extremely intricate, 8k, ..."

Prompt: "Photorealistic blonde girl in pyjama."

Prompt: "best sneakers 1d539 abccb NIKEiD LeBron Soldier 12 Designs | Sole Collector."

Prompt: "ROCK & REPUBLIC 'Neil' Relaxed Straight Leg Jeans, Main, color, 490."

Prompt: "Java Ruched Faux Solid Taffeta Curtain."

Prompt: "Chuck Taylor All Star Hi W."

PeRFlow-XL   Rectified Diffusion-XL   PeRFlow-XL   Rectified Diffusion-XL

Figure 10: Qualitative comparison.

