# OpenReview forum: "Rectified Diffusion: Straightness Is Not Your Need in Rectified Flow"
_ICLR.cc/2025/Conference — ICLR 2025 Poster_

### Official Review · Reviewer_2rVP · 2024-10-24

**Soundness:** 3
**Presentation:** 3
**Contribution:** 3
**Rating:** 8
**Confidence:** 4

**Summary:**

This paper proposes rectified diffusion, a generalization of rectified flow to first-order ODEs. During training, rectified diffusion aims to correct the learned ODE of the pretrained diffusion model to be closer to a first-order ODE, using precomputed noise-data pairs. Thus the authors argue that using paired noise-data examples is the essential component of both rectified flow and rectified diffusion. The authors further propose phased rectified diffusion and consistency distillation on top of rectified diffusion. In their experiments, they demonstrate superior quantitative and qualitative performance over rectified flow and a previous phased ODE segmentation method (PeRFlow).

**Strengths:**

* The quantitative and qualitative results (Figs. 8-10) are quite convincing. The authors provide the essential baselines of rectified flow and PeRFlow, as well as additional Stable Diffusion XL-based baselines in Table 2.
* The idea is original yet simple (a strength not a weakness). It is well-backed by a brief theoretical discussion in Section 2.2.
* Besides a few grammatical errors and too-short figure captions, the presentation is good. The text is easy to follow, and the figures are intuitive.

**Weaknesses:**

* It would have been nice to compare to Consistency Models (Song et al. 2023) for one-step generation.
* In Table 1, is there a typo for 25-step Rectified Diffusion (Ours)? It says the time is only 0.09s, which is the same as 1-step Rectified Diffusion (Ours).

Nitpick:
* Figure 4 should have axis labels.
* Fonts too small to read in Figure 7.
* Figure 6 caption too short and could use more description.

**Questions:**

* In lines 357-358, the authors claim that the first-order ODE objective is a stronger constraint than self-consistency. If it’s the case that first-order ODE => self-consistency, then why would consistency distillation be necessary after rectified flow?

---

> ### Author Response · Authors · 2024-11-23
> **[1/1]**
>
> Thank you for your encouraging review!
>
>
>
> **W1** "It would have been nice to compare to Consistency Models (Song et al. 2023) for one-step generation."
>
>
>
> Our work and compared works are built upon Stable Diffusion for text-to-image generation. The original works of Consistency Models [1] is only suitable for unconditional and class-conditional generation. We reported the performance comparison with Latent Consistency Models [2] . Please refer to Line 455, lines 478-479 in Table 2.
>
>
>
> **W2 & 3** "Typos and revision advice."
>
> Thank you for your careful reading and constructive advice. We will revise our paper following your suggestion.
>
>
>
> **Q1** "In lines 357-358, the authors claim that the first-order ODE objective is a stronger constraint than self-consistency. If it’s the case that first-order ODE => self-consistency, then why would consistency distillation be necessary after rectified flow?"
>
>
>
> **To set up fair comparison.** The most direct reason why we adopted distillation to further improve performance is to set up a fair comparison with [3]. The difference is that we adopt consistency distillation instead of the naive distillation strategy adopted in [3]. We achieve better 1-step performance than the distilled baseline in [3] with only 3% GPU days used by [3] for distillation. This is also a default procedure proposed in the original work of rectified flow (page 8 of [4]).
>
>
>
> **Rectification is a harder objective than distillation.** The root reason is that first-order ODE is a harder target than distillation.  Intuitively speaking, since the first-order ODE objective is a stronger constraint than self-consistency, it is harder to train and will face more optimization issues ([The optimization issue is also revealed in [5]).
>
>
>
> The crucial difference between rectification (aims to achieve first-order ODE) and distillation (aims to directly predict the $\mathbf x_0$) is that rectification will change the ODE trajectory but distillation does not.  We provided a visualization explanation in Figure 5.  Since it’s hard to intuitively determine whether a curved ODE path satisfies first-order linearity, we represent the first-order ODE path with a straight line (It is not straight in many cases). Consistency models change the prediction of original diffusion models but do not change the ODE trajectory. It makes it only suitable for stochastic multistep sampling (If it uses diffusion samplers like DDIM, it will fall out from the ODE trajectory, causing the further prediction to be degraded). Instead, rectification smoothens and rectifies the ODE trajectory to make it the first-order ODE.  Generally, after rectification as introduced in our work, the diffusion model is still a diffusion model (since its prediction direction still follows the derivative of its ODE as shown in Figure 5) (The contrast part is that diffusion models after consistency distillation will become consistency models) but performs much better in few-step setting. The high flexibility required by first-order ODE for inference requires more challenges in training.
>
>
>
> However, despite the difficulty of achieving perfect first-order ODE, after rectification,  the ODE of our model will become generally first-order approximate. It will reduce the difficulty of further distillation if we use the model that has undergone rectification as the teacher model for distillation. (This is also revealed in [4] for the specific case of rectified flow forms.)
>
>
>
> Hope our explanation makes the the difference and root reason clear to you.
>
> Sincerely,
>
>
>
> [1] Consistency Models. ICML 2023.
>
> [2] Latent Consistency Models: Synthesizing High-Resolution Images with Few-Step Inference.
>
> [3] InstaFlow: One Step is Enough for High-Quality Diffusion-Based Text-to-Image Generation. ICLR 2024.
>
> [4] Flow Straight and Fast: Learning to Generate and Transfer Data with Rectified Flow. ICLR 2023.
>
> [5] Improving the Training of Rectified Flows. NeurIPS 2024.

---

> > ### Comment · Reviewer_2rVP · 2024-11-26
> >
> > Thank you to the authors for their response. I will keep my score of 8.

---

> ### Author Response · Authors · 2024-11-26
> **Thank you sincerely for the very positive review!**
>
> Thank you for maintaining the positive score. We sincerely appreciate the time and effort you invested in reviewing our paper. Your feedback has been exceptionally valuable, and we are deeply grateful for your support.
>
> We are always here should you have any further questions or need additional information.
>
> Best regards,
>
> The Authors

---

### Official Review · Reviewer_t4qC · 2024-10-24

**Soundness:** 1
**Presentation:** 2
**Contribution:** 1
**Rating:** 1
**Confidence:** 4

**Summary:**

This paper proposed to study the essence of rectified flow and proposed Rectified Diffusion.

**Strengths:**

They tried to highlight that straightness is not an essential training target for rectification.

**Weaknesses:**

It was not stated clearly why it is necessary to study whether straightness is not an essential training target for rectification. To me, it was the straightness used in the ode of the rectified flow algorithm that causes the usage of rectified flow, thus it seems trivial that there is no need to use the rectified flow in a ode that does not follow a straight line.

The diffusion that follows the curve form that the authors proposed is mentioned in some papers including the original rectified flow paper with the formula listed as well. It can return back to DDPM or DDIM; therefore, to me, it seems like recirculating the algorithm.  The authors might want to look into the following references mentioned in the original rectified flow paper in Lu et al., 2022 (Lipman et al., 2022; Albergo & Vanden-Eijnden, 2022; Heitz et al., 2023).Neklyudov et al. (2022).
While the authors meant to straighten the paths that originally were curves, they can be viewed as approximately meanwhile they can also be viewed as the original straight flows.
Although one can choose any ode path, the paper did not prove (or it is not clear) that their choice gains advantage over existing algorithms. As a matter of fact, the authors used 200000 runs to train the algorithm as shown in figure 2.

Also, Theorem 1 proved in Appendix seems like a similar version of one in (Lu et al., 2022), there is no need to repeat.

As a result of the weaknesses stated here, the title of this paper seems like questionable itself ( seems like a logic flaw?).

**Questions:**

1. Please correct the typo "obeserve" in Page 5 line 265 and if there is any more.
 2. In Page 5 around line 265, the authors stated that the trajectory of y_t is a straight line from the initial point ϵ towards the direction of x0 (as shown in Figure 3 (e). This is not clearly explained. Also, It seems like a very special case  when they only used X0 = [0, 1]. It would make more sense if the authors generate the trajectories over time for each elements of X_0 to \epsilon.
3. In Page 4, line 205, the authors mentioned that there exists an exact ODE solution form from Lu et al., 2022, but they omitted the solution was  for  what first order ODE. That will easily confuse the readers.
4. Eqn (2)  in Page 4 should be just an linear approximation of curves. You can use a straight line to approximate the curve, but then you can say this linear line is the result of a straight flow ode.
5. Some descriptions seem like claims that need to be further elaborated, for example those in line 250-line 261.

---

> ### Author Response · Authors · 2024-11-23
> **[1/5]**
>
> Thank you for your comment. We understand there is a **considerable misalignment**. We value the chance to elaborate more on Rectified Diffusion.
>
> We try our best to give a detailed, self-contained rebuttal and hope it can clarify the misalignment. Please kindly read our detailed rebuttal.
>
>
>
> At the beginning, we provide a section of background and motivation summary of our research before answering specific questions.
>
> **Summary of background and motivation in our research**
>
> There are a lot of off-the-shelf well pretrained diffusion models. They follow a general diffusion form $\mathbf x_t= \alpha_t \mathbf x_0 + \sigma_t \boldsymbol \epsilon$ [1]. For example, Stable Diffusion [11], which follows the DDPM form (That is, the diffusion form is defined as $\mathbf x_t = \alpha_t \mathbf x_0 + \sigma_t \boldsymbol \epsilon $, where $\alpha_t ^ 2+ \sigma_t ^ 2 = 1$). EDM [10], which follows the VE form (That is, the diffusion form is defined as $\mathbf x_t = \mathbf x_0 + \sigma_t \boldsymbol \epsilon$, where $\alpha_t = 1$, $\sigma_t$ can be $[0, \sigma_\max]$).  For rectified flow, it is equivalent to setting $\alpha_t = 1-t$ and $\sigma_t = t$.
>
>
>
> However, diffusion models are known for slow generation due to the iterative denoising process. Rectified Flow proposed the technique termed as rectification (reflow) which is a progressive retraining technique that can significantly improve the sample quality in few-step sampling while still maintaining the flexibility for more steps inference.
>
>
>
> However, the rectification is considered to be only applicable to rectified flow-based diffusion forms (that is we must have $\mathbf x_t = (1-t) \mathbf x_0 + t \boldsymbol \epsilon $ for retraining in the rectification process.) in prior research [2-5]. Straightness is thought to be the reason why it is only suitable for diffusion models with rectified flow forms to enjoy the benefits of rectification for faster sampling.
>
>
>
> For example, Stable Diffusion is currently one of the most widely used diffusion models. It was originally trained with DDPM form (i.e., $\mathbf x_t = \alpha_t \mathbf x_0 + \sigma_t\boldsymbol \epsilon$, $\alpha_t^2 + \sigma_t ^2 = 1$) and $\boldsymbol \epsilon$-prediction.  However, to improve its generation quality in the few-step setting, [3] initializes with the pretrained weight of Stable Diffusion but retrains it with rectified flow ($\alpha_t = 1-t, \sigma_t =t$) form and $\mathcal v$-prediction.
>
>
>
>  This coversion (from DDPM to Rectified Flow, from $\boldsymbol \epsilon$-prediction to $\boldsymbol v$-prediction) makes the overall procedure to adopt rectification for sampling acceleration cumbersome. It also causes a large gap between pretaining and further retraining in rectification.
>
>
>
> Our research is to say:
>
>
>
> - The conversion to rectified flow is unnecessary and can harm the performance.
> - Rectification is suitable for general diffusion models instead of only applicable form rectified flow forms.
> - The essential training target is not to achieve straight ODE but first-order ODE. First-order ODE can be curved in essence.

---

> ### Author Response · Authors · 2024-11-23
> **[2/5]**
>
> **W1** "It was not stated clearly why it is necessary to study whether straightness is not an essential training target for rectification."
>
> Because this makes our method **simpler, stronger, and more general.**
>
>
>
> **Simpler**
>
> We do not need the conversion from the pretrained diffusion forms into rectified flow forms, which greatly simplifies the overall training procedure (Abstract, Figure 1). We keep everything of the pretrained diffusion models unchanged, including noise schedulers, prediction types, network architectures, and even training and inference code. The only difference is that the noise and data adopted for training are pairs generated by pretrained diffusion model. (lines 90-94)
>
> **Stronger**
>
> We have no gap in prediction type and diffusion form gap between the pretraining and retraining (rectification). We keep everything of the pretrained diffusion models unchanged, including noise schedulers, prediction types, network architectures, and even training and inference code. (lines 90-94). This allows for great improvements in training efficiency. In Figure 2, we show that our one-step performance significantly surpasses the one-step performance the one-step performance of the official weight of the Rectified Flow method [3] within 20,000 iterations. It takes only 8% number of  trained images adopted for training the official weight [3].
>
> **More general**
>
> We show that rectification is suitable for general diffusion forms, which extends the scope of rectification for general diffusion forms.
>
>
>
>
>
>
>
> **W2** "To me, it was the straightness used in the ode of the rectified flow algorithm that causes the usage of rectified flow, thus it seems trivial that there is no need to use the rectified flow in a ode that does not follow a straight line."
>
>
>
> We hope to clarify that **the ODE of rectified flow is not straight** if  **rectification** is not conducted [2-5].  Therefore it might be improper to say "straightness used in the ode of the rectified flow  that causes the usage of rectified flow".
>
> Using rectified flow does not mean straight ODE path, especially when the rectification is not well conducted.  This is because $\boldsymbol v_{t} = \mathbb E [\mathbf x_0 - \boldsymbol \epsilon|\mathbf x_t]$ can be various value if $\mathbf x_0$ and $\boldsymbol \epsilon$ is not well paired for retraining.  However, even though the ODE is not straight (the rectification is not conducted), many works are using rectified flow forms for training and inference [6-7].
>
>
>
>
>
> **W3** "The diffusion that follows the curve form that the authors proposed is mentioned in some papers including the original rectified flow paper with the formula listed as well.  To me, it seems like recirculating the algorithm"
>
> All our listed diffusion forms are common diffusion forms discussed and implemented in many previous works [9-10]. We are not proposing new diffusion forms but to show that general diffusion forms can adopt rectification for diffusion acceleration.  We are not recirculating algorithms.
>
>
>
> **W4** "While the authors meant to straighten the paths that originally were curves, they can be viewed as approximately meanwhile they can also be viewed as the original straight flows."
>
>
>
> **Your question seems to be unclear**.  "they can be viewed as approximately" lacks object which seems to be an incomplete sentence, and “meanwhile” is grammatically misplaced.  We rephrase it as follows and hope it follows your original question:
>
> "While the authors meant to straighten the paths that were originally curves, meanwhile they can also be viewed as the original straight flows."
>
> Firstly, we hope to clarify **we are never meant to straighten the paths that originally were curved**.  We are not meant to straighten ODE or sampling through straight flows, which is consistently highlighted in our title,  abstract (lines 22-23), introduction (lines 95-102), analysis (lines 262-269),  Figure 1, Figure 3. In these areas, we repeatedly emphasized that "the first-order ODE path for diffusion models like DDPM and Sub-VP is inherently curved instead of straight."  Therefore, we never try to use straight lines to approximate curves. Instead, our target is to rectify the original ODE into a first-order ODE. The first-order ODE can be curved (as illustrated in Figure 3). This is consistent with our title that claiming straightness is not your need.
>
>
>
> Secondly, **original flows are not straight either.** Since you have assumed that the paths were **original curves** in the first sentence "**paths that were originally curves**", it might be improper to say "They can be viewed as the **original straight flows**."

---

> ### Author Response · Authors · 2024-11-23
> **[3/5]**
>
> **W5** "The paper did not prove (or it is not clear) that their choice gains advantage over existing algorithms. As a matter of fact, the authors used 200000 runs to train the algorithm as shown in figure 2."
>
>
>
> All our experimental results **consistently show the performance gains** of Rectified Diffusion over Rectified Flow-based methods.
>
>
>
> **Rectified Diffusion  outperforms Rectified Flow with significantly smaller training costs.**   As highlighted the Figure 2, where we show the performance change as the training iterations. Our one-step performance significantly outperforms the one-step performance of the official weight of Rectified Flow within only 20, 000 iterations with batch size 128. The overall number of trained images at this iteration is only 8% of the number of trained images in Rectified Flow (lines 498-499, lines 123-125).   Even after the full training of 200, 000 iterations, considering that our batch size adopted is only 128 which is  significantly smaller than the batch size adopted in previous work [3], we still use less trained images than the official weight of Rectified Flow method (lines 427-493).
>
>
>
> **Rectified Diffusion significantly outperforms Rectified Flow in few-step setting.** The performance in few-step setting is the most important criterion to judge the effectiveness of rectification on the ODE (since we aim to achieve efficient sampling and improve the performance in few-step sampling). Rectified Diffusion significantly outperforms Rectified Flow in this setting. Specifically, on COCO-5K. Rectified Flow achieves one-step FID 47.91, two-step FID 31.35. Instead, Rectified Diffusion achieves one-step FID 27.26 (significantly outperforms the two-step FID of Rectified Flow), two-step FID 22.98 (approaching the 25-step FID of Rectified Flow 21.65). The is strong evidence that Rectified Diffusion is better than Rectified Flow.
>
>
>
> **Rectified Diffusion has better performance upper bound than Rectified Flow.** It should be noted that Rectified Diffusion and Rectified Flow are all trained with generated data from pretrained diffusion models. Specifically, in the default setting, they both use 25-step DPMSolver [4] with Stable Diffusion v1-5  to generate noise-data pairs. Therefore, the performance of 25-step DPMSolver with the teacher diffusion model works as the performance upper-bound. The 25-step Rectified Diffusion achieves better performance than  25-step Rectified Flow on both FID and CLIP SCORE. Therefore, Rectified Diffusion has better performance upper bound than Rectified Flow. Additionally, please note that the 25-step performance of Rectified Diffusion and that of Rectified Flow are already very close to the 25-step DPMSolver with the teacher diffusion model. Therefore, we can not achieve a very large improvement.
>
>
>
> **Rectified Diffusion (Phased) consistently surpasses the baseline PeRFlow.**  We extend Rectified Diffusion into the phased setting, our performance still consistently outperforms the previous phased rectified flow-based method.
>
>
>
> **Rectified Diffusion (CD) surpasses the baseline Rectified Flow (Distill) with only 3% GPU days for training.**   The distillation process described in the InstaFlow paper takes 110 A100 GPU days. Our training cost is approximately 3% of the GPU days of InstaFlow’s distillation process. Moreover, the results generated by Rectified Flow (Distill) remain relatively blurry and lack the ability for multi-step refinement, which limits its applicability. Instead, Rectified Diffusion (CD) not only achieves better one-step performance but also supports multistep refinement to further improve the performance.
>
>
>
> Therefore, we believe our validation and empirical evidence are convincing and sufficient.
>
>
>
> **W6**  "Theorem 1 proved in Appendix seems like a similar version of one in (Lu et al., 2022), there is no need to repeat."
>
>
>
> Our theorem 1 is to show the equivalence between Equation 1 and Equation 2 holds if and only if the epsilon prediction is constant. Equation 1 is an important conclusion from DPM-Solver, which works as the basis of our proof of theorem 1. However, our theorem 1 is different from proofs in DPM-Solver with entirely different question, motivation, and proof procedure.

---

> ### Author Response · Authors · 2024-11-23
> **[4/5]**
>
> **Q1** Typo.
>
> Thank you for pointing it out. We will correct it.
>
>
>
> **Q2.1** "On page 5 around line 265, the authors stated that the trajectory of y_t is a straight line from the initial point ϵ towards the direction of x0 (as shown in Figure 3 (e). This is not clearly explained. "
>
> $\mathbf y_t$ is defined as $\mathbf y_t := \frac{\alpha_t}{\sigma_t} \mathbf x_0 + \boldsymbol \epsilon$ where $\mathbf x_0$ and $\boldsymbol \epsilon$ is the sample nosie pair.  $\gamma_t = \frac{\alpha_t}{\sigma_t}$ is a monotonically decreasing function with respect to time t with range space $[0, \gamma_\max]$ (Please see Equation 2 and the following lines in page 4 of [1]).
>
> Note that $\gamma_t$ is the only changeable factor along the time axis, and both $\mathbf x_0$ and $\boldsymbol \epsilon$ are constants.
>
> Therefore, it is equivalent to the simplest form of the linear function $y = ax +b$, where $y := \mathbf y_t$ , $a:=\mathbf x_0$, $b:=\boldsymbol \epsilon$, and $x:=\gamma_t$ . This indicates that the point will move from the point $b:=\boldsymbol \epsilon$ following the direction $a:=\mathbf x_0$
>
> From a different perspective,  the derivative of $\frac{d \mathbf y_t}{dt} = \frac{d\gamma_t}{dt} \mathbf x_0 $. $\frac{d\gamma_t}{dt}$ is a scalar function (scalar does not influence the direction of a vector). Therefore the direction of the derivative is always the same as the direction of $\mathbf x_0$.
>
> We believe this should be easily observed with minor exposure to linear algebra.  Hope our explanation can clarify your confusion.
>
>
>
> **Q2.2**  Also, It seems like a very special case when they only used X0 = [0, 1]. It would make more sense if the authors generate the trajectories over time for each elements of X_0 to \epsilon.
>
> It is not a special case. Figure 3 (e) is exactly generated over time through the definition.  Please refer to Q2.1.
>
>
>
> **Q3** On page 4, line 205, the authors mentioned that there exists an exact ODE solution form from Lu et al., 2022, but they omitted the solution was for what first order ODE. That will easily confuse the readers.
>
>
>
> We kindly correct you that **the exact solution proposed by Lu et al., 2022 is not for first-order ODE**. The function describes the general ODE. First-order ODE is a special case of the general ODE. We provided the definition of first-order ODE in Equation 2.
>
>
>
> **Q4**  "Eqn (2) in Page 4 should be just an linear approximation of curves. You can use a straight line to approximate the curve, but then you can say this linear line is the result of a straight flow ode."
>
>
>
> Firstly, please note that there is a fundamental difference between linear approximation of curves and first-order ODE.
>
> Linear approximation: When the ODE trajectory does not satisfy first-order property, diffusion models can generate data by splitting the ODE into consecutive small intervals and adopting the DDIM solvers for numerical solving. In this case, the intervals should be small enough to achieve a tolerable error in sampling. When adopting this idea for fast sampling such as one-step generation, diffusion models cannot generate correct samples. This is because the first-order approximation error is too large.
>
>
>
> First-order ODE:  When the ODE trajectory is a first-order property, diffusion models can sample high-quality data with arbitrary inference steps. We do not need the interval of each step (step size) to be small, and the step size can be arbitrarily large without causing any approximation error to the ODE. When the step size is equivalent to the length of time axis (i.e., T), we can enable one-step generation.
>
>
>
> Secondly, we hope to re-emphasize that we are not meant to straighten ODE or sampling through straight flows, which is consistently highlighted in our title,  abstract (lines 22-23), introduction (lines 95-102), analysis (lines 262-269),  Figure 1, Figure 3. In these areas, we repeatedly emphasised that "the first-order ODE path for diffusion models like DDPM and Sub-VP is inherently curved instead of straight."  Therefore, we never try to use straight line to approximate the curve. Instead, our aim is to rectify the original ODE into a first-order ODE. The first-order ODE can be curved (as illustrated in Figure 3). This is consistent with our title that claiming straightness is not what we need.
>
>
>
> Thirdly, we kindly remind you that the **ODE trajectory should not be viewed from a discretized perspective**. It is a concept in continuous time. The discretized trajectory with loose discretization can not reflect the actual trajectory of the ODE.

---

> ### Author Response · Authors · 2024-11-23
> **[5/5]**
>
> **Q5** "Some descriptions seem like claims that need to be further elaborated, for example those in line 250-line 261."
>
>
>
> The reason why the ODE path does not follow the first-order property is due to the random noise-data pair adopted during training. In the specific case, the reason why the ODE path of rectified flow is not straight is also due to the random noise-data pair adopted during training [2-5].
>
> For example, for a diffusion model trained with the form $\mathbf x_t = \alpha_t \mathbf x_0 + \sigma_t \boldsymbol \epsilon$. The score $\boldsymbol s_{\boldsymbol \theta}$ will converge to the expectation value of all the possible conditional scores $\nabla_{\mathbf x_t} \log \mathbb P(x_t|\mathbf x_0) $ due to the noise $\boldsymbol \epsilon$ and data $\mathbf x_0$ is randomly paired [9]. That is
> $$
> \boldsymbol s_{\boldsymbol \theta} (\mathbf x_t , t) = \mathbb E_{\mathbb P(\mathbf x_0 \mid \mathbf x_t)} [\nabla_{\mathbf x_t} \log \mathbb P(\mathbf x_t|\mathbf x_0) ].
> $$
> Generally speaking, the expectation score value will converge to different values and directions [2-5, 9]. Therefore, on the PF-ODE, the score $\boldsymbol s_{\boldsymbol \theta}(\mathbf x_t, t)$ will change along the time axis. Since $\boldsymbol \epsilon = - \sigma_t \boldsymbol s_{\boldsymbol \theta}$ , it means that the epsilon prediction will change along the time axis, which is not comprised with our theorem 1. It is not a first-order ODE.
>
>
>
> However, if we achieve perfect noise data coupling and satisfy the no-intersection condition, $\mathbf x_t$ will be sampled with a single noise and data pair.  Without optimization errors, we will have
> $$
> \boldsymbol s_{\boldsymbol \theta} (\mathbf x_t, t) = \nabla_{\mathbf x_t} \log \mathbb P(\mathbf x_t \mid \mathbf x_0 ) = -\frac{\mathbf x_t - \alpha_t \mathbf x_0}{\sigma_t^2} = -\frac{\boldsymbol \epsilon}{\sigma_t},
> $$
> Since $\boldsymbol \epsilon_{\boldsymbol \theta} = - \sigma_t \boldsymbol s_{\boldsymbol \theta}$ , therefore the epsilon prediction on the PF-ODE should always be the $\boldsymbol \epsilon$.  That is, we achieve the first-order ODE.
>
>
>
>
>
>
>
> [1]  Variational Diffusion Models. NeurIPS 2021.
>
> [2] Flow Straight and Fast: Learning to Generate and Transfer Data with Rectified Flow. ICLR 2023.
>
> [3]  InstaFlow: One Step is Enough for High-Quality Diffusion-Based Text-to-Image Generation. ICLR 2024.
>
> [4] Improving the Training of Rectified Flows. NeurIPS 2024.
>
> [5] PeRFlow: Piecewise Rectified Flow as Universal Plug-and-Play Accelerator. NeurIPS 2024.
>
> [6] Scaling Rectified Flow Transformers for High-Resolution Image Synthesis. ICML 2024.
>
> [7] Pyramidal flow matching for efficient video generative modeling.
>
> [9] Score-Based Generative Modeling through Stochastic Differential Equations. ICLR 2021 Oral.
>
> [10] Elucidating the Design Space of Diffusion-Based Generative Models. NeurIPS 2022.
>
> [11] High-Resolution Image Synthesis with Latent Diffusion Models. CVPR 2022.

---

> ### Author Response · Authors · 2024-11-25
> **Reminder for response: Discussion Period Ending Soon**
>
> Dear Reviewer t4qC,
>
> As the discussion deadline approaches within 48 hours, we kindly ask if our response has addressed your concerns. Please feel free to share any additional questions or feedback, and we’ll be happy to provide further clarification.
>
> Best regards,
>
> The Authors

---

> ### Author Response · Authors · 2024-11-29
> **Reminder for response: We are looking forward to your reply**
>
> Dear Reviewer t4qC,
>
> We kindly ask if our response has addressed your concerns. Please feel free to share any additional questions or feedback, and we’ll be happy to provide further clarification.
>
> Best regards,
>
> The Authors

---

> > ### Comment · Reviewer_t4qC · 2024-11-30
> >
> > I thank the authors for their lengthy reply to my concerns.  Like me, most reviewers were initially confused about why the authors presented a straight path in one graph, in this aspect, I appreciate the authors revised their paper, which helped me understand their algorithm better.
> >
> > Their response shown that they only experimentally demonstrated that their rectified diffusion achieves "performance gains". I am concerned if this is only for specific datasets.
> >
> > In the algorithm that the authors proposed, they started from  in eqn. (1), which is the exact solution in (Lu et al., 2022). Somehow, the authors claimed that "the exact solution proposed by Lu et al., 2022 is not for first-order ODE. The function describes the general ODE". I disagree. First order ODE u′ = f(t, u) can be a quite general form. I don't see any reason to exclude Lu's solution as for the first order ODE. The author further stated that "we provided the definition of first-order ODE in Equation 2".  To arrive to this equation, they took advantage of the assumption that the epsilon is a constant.  However, this is typically not true if we look at original diffusion model in most of the research papers. Setting the epsilon as a constant indeed is an estimation. Therefore, we have many existing contributions, including DPM solver, on developing fast algorithms to predict the trajectory of the reverse diffusion to reflect the forward trajectory; although this paper claimed that they can recover the original x_0, this is based on eqn. (2), which should be viewed as an approximation of the exact solution in eqn. (1). It was the different noise added over the time trajectory, t, that makes it difficult for the reverse sampling, which indeed is also the advantage of diffusion model. Thus, I view figure 4e as misleading.
> >
> > Regarding Theorem 1, the authors claimed that: "our theorem 1 is different from proofs in DPM-Solver with entirely different question, motivation, and proof procedure". This seems overstated. Thus, I took a more careful look at their proof on Theorem 1, and noticed that it is just a simply calculation on the formula in Eqn. (1) (Lu 2022) by pulling out the epsilon as a constant from the integral, which I believe a beginner from an intro calculus class can do; I am sure most readers can see this. I would appreciate if the authors spend their precious space in this paper to provide a proof on the advantage that they claimed.

---

> ### Author Response · Authors · 2024-11-30
> **Thank you for your reply.**
>
> Thank you for your reply. We tackle your concerns as follows:
>
>
> > 1. Is the validation only for specific datasets?
>
> No, we have **already provided additional validation** on Laion and CC12M in addition to COCO-2017 and COCO-2014.
>
> > 2. The difference between the general ODE solution and our presented first-order ODE solution.
>
> Our presented first-order ODE solution is a specific form of the general ODE solution. It has a great property that supports efficient sampling. Our method transforms the general ODE into first-order ODE through rectification instead of approximation.
>
> **Contrary to your comment, first-order ODE will not suffer from any level of approximation error even when first-order solvers are used for sampling.**
>
> > 3. Assume $\epsilon$ as constant.
>
> We are not assuming $\epsilon$ as constant, which seems to be significantly misleading of our method. Instead, we show that the $\epsilon$ on the same PF-ODE will become constant as long as we achieve the first-order property. Our motivation is to retrain the diffusion model to satisfy the first-order property instead of using first-order approximation for approximate sampling, which we have **already emphasized in our reply to Q4**.
>
> > 4. The difference between Theorem 1 and DPM-Solver.
>
> 4. 1 Question:
>
> - DPM-Solver is for **training-free** inference sampling.
>
> - Rectified Diffusion is for **retraining** to accelerate sampling.
>
> 4. 2 Motivation:
>
> - DPM-Solver is to use low-order solvers to achieve a better approximate of PF-ODE.
>
> - Rectified Diffusion is to rectify the PF-ODE into a smoother first-order ODE to support efficient sampling through retraining.
>
>
> 4. 3 Proof:
>
> - DPM-Solver adopts Taylor expansion for low-order approximation of the integral.
>
> - Our proof shows the first-order property satisfies if and only if the $\boldsymbol \epsilon$ is constant on the PF-ODE.
>
>
> Please let us know if our response has addressed your concerns. We look forward to your further reply!
>
> Best,
>
> Authors

---

> > ### Comment · Reviewer_t4qC · 2024-11-30
> >
> > Thank you for your response! I will keep my score.

---

> > > ### Author Response · Authors · 2024-11-30
> > > **We kindly correct you that the ODE should not be viewed from the discretized perspective.**
> > >
> > > For your comment "You can use a straight line to approximate the curve, but then you can say this linear line is the result of a straight flow ode.", which seems to be an improper understanding of the concept ODE.  ODE is a concept in continuous time. We can only approximate the ODE when the discretization interval $\Delta t \rightarrow 0$ instead of using very loose discretization to represent the ODE ($\Delta t$ is larger).

---

> > > ### Author Response · Authors · 2024-11-30
> > > **We kindly correct you we never meant to straighten ODE or sampling through straight flows.**
> > >
> > > For your comment "The authors meant to straighten the paths that originally were curves", which seems to be a significant misunderstanding of our work.
> > >
> > > The discussion between the straight ODE path and curved ODE path is exactly one of the most important motivations of our work.   Specifically, we never meant to straighten ODE or sampling through straight flows, which is **consistently highlighted in our title, abstract (lines 22-23), introduction (lines 95-102), analysis (lines 262-269), Figure 1, Figure 3.** In these areas, we repeatedly emphasized that "the first-order ODE path for diffusion models like DDPM and Sub-VP is inherently curved instead of straight." Therefore, we never try to use straight lines to approximate curves. Instead, we aim to rectify the original ODE into a first-order ODE. The first-order ODE can be curved in most cases (as illustrated in Figure 3).

---

> ### Author Response · Authors · 2024-11-30
> **We kindly correct you that using low-order solvers for low-order approximation/estimation sampling is inherently different from rectifying the ODE to satisfy the first-order property.**
>
> Following the proof of DPM-Solver-1 (Equation 3.6 and the following Equation in Page 5 of DPM-Solver) which is also equivalent to DDIM sampler, we can approximate the ODE solution from $t$ to $s$ as following:
>
> $$
> \mathbf x_s = \frac{\alpha_s}{\alpha_t}\mathbf x_t - \alpha_s \int_{\lambda_t}^{\lambda_s}e^{-\lambda} \boldsymbol \epsilon(\mathbf x_{t_{\lambda}}, t_{\lambda})  \mathrm d \lambda  + \mathcal O((\lambda_s - \lambda_r)^2).
> $$
>
> In the DPM-Solver-1, they just ignore the later high-order errors $ \mathcal O((\lambda_s - \lambda_r)^2)$ for inference, which is invalid when $\lambda_s - \lambda_t$ is large. **This is also the direct reason why DPM-Solver-1 (DDIM) needs at least 50 steps for high-quality generation.**
>
>
> In our paper, we are not proposing any level approximation. Our motivation is to rectify the PF-ODE to satisfy the first-order property, **which will eliminate the later high-order error terms** through paired retraining (Rectification). This is the fundamental reason why **Rectified Diffusion can support 1-step generation**. On the contrary, DPM-Solver needs at least over ten steps for high-quality generation.

---

### Official Review · Reviewer_KrPe · 2024-10-26

**Soundness:** 3
**Presentation:** 3
**Contribution:** 3
**Rating:** 6
**Confidence:** 3

**Summary:**

This paper presents two findings, one is that straightness as required in rectified flow is not necessary, and second is that it proposed Rectified Diffusion, which generalizes the design space and application scope of rectification to diffusion models.

**Strengths:**

First strength is that the paper shows that straightness is necessary as required in rectified flow. It shows that rectified flow is a special case of rectified diffusion, which is what the paper proposed as a better generative model.
Second strength is the better quantitative results and favorable comparison with other methods.

**Weaknesses:**

Maybe I am missing something, it seems the paper argues that straightness of the ODE trajectory is not necessary, but its results appear to show a straight path, such as Figure 4 and Figure 5.
Other weaknesses include that from visual evaluation, it seems that though the rectified diffusion gave better images than rectified flow in most cases, the improvement is not always clear. For example, in Figure 10, the result of rectified diffusion on the sneakers seem that the string missed a buttonhole. Also, in the pajama example, figure patterns on the pajama in the rectified diffusion result are not as sharp as those on the result of rectified flow. In Figure 10, the huskie images generated by both ReFlow and Rec Diffusion seem to have an extra leg, though it is more evident in the image generated by ReFlow.

**Questions:**

From eq 13 to eq 14, it asserts that d\lambda_t/dt is not zero, what is that? Or as \lambda_t = ln \alpha_t / \sigma_t, is it always true that this partial derivative is non-zero?
Table 1, second block, it seems for example Rectified Flow's FID had great improvement from 1 step to 2 steps then to 25 steps, the Rectified Diffusion's FID did not show a great improvement from 1 step to 2 steps then to 25 steps, while the Rectified Diffusion's FID at 25 steps is slightly better than the FID of Rectified Flow at 25 steps. Can authors elaborate on why Rectified Diffusion does not experience a much-improved performance from 1 step to 25 steps?
Figure 2 is somehow difficult to understand. Why did the FID and CLIP score of Rectified Flow not change with iterations?
Figure 6, the two rightmost plots, why did the curves of Rectified Diffusion go up at 4 and 8 steps? Does it mean the method need to stop at some optimal points before its performance decreases? And if so, how does one determine the proper steps to stop?
In Figure 7, it seems the biggest difference in preference given by human evaluation is the Phased 4-step results. Does it imply that Phased implementation, which features multiple straight trajectories rather than one single straight trajectory generally, generally gives better results? Also, in Phased implementation, how many phases were used? It seems this information is not given in the paper.

---

> ### Author Response · Authors · 2024-11-23
> **[1/2]**
>
> Thank you for your in-depth review.
>
>
>
> **W1** "Maybe I am missing something, it seems the paper argues that straightness of the ODE trajectory is not necessary, but its results appear to show a straight path, such as Figure 4 and Figure 5."
>
> Please kindly refer to lines 294-297 and the caption of Figure 4 and Figure 5, where we stated that since it is hard to visually tell if a curved ODE satisfies first-order, we adopt the straight line for clearer demonstration.
>
> For your convenience, we attach the sentences stated in our paper in the following:
>
> "Since it’s hard to intuitively determine whether a curved ODE path satisfies first-order linearity, we represent the first-order ODE path with a straight line."
>
> "We apply straight lines for clearer demonstration."
>
> We will make the statement clearer in the revision.
>
>
>
> **W2** "Visual evaluation, it seems that though the rectified diffusion gave better images than rectified flow in most cases, the improvement is not always clear. "
>
> Since diffusion models transform random noise into samples, the generation is stochastic in nature considering the randomness of the initial noise. Therefore, although our model has better average performance in that we can say our model is more likely to produce better samples, we can not ensure that our method always produces better samples than other models.
>
>
>
> **Q1** "From eq 13 to eq 14, it asserts that d\lambda_t/dt is not zero, what is that? Or as \lambda_t = ln \alpha_t / \sigma_t, is it always true that this partial derivative is non-zero"
>
> As stated in lines 149-156, we adopt the general diffusion form $\mathbf x_t = \alpha_t \mathbf x_0 + \sigma_t \boldsymbol \epsilon$ for discussion. This definition follows previous works [1] [2].  Following the definition in [1] (Please see Equation 2 and the following lines on page 4 of [1]), $\frac{\alpha_t}{\sigma_t}$ should be strictly monotonically decreasing in time.
>
> Therefore, $\lambda_t = \ln \frac{\alpha_t}{\sigma_t}$ should be strictly monotonically decreasing in time. We have the $\frac{d \lambda_t}{d_t} < 0 $.
>
> **Q2** "Can authors elaborate on why Rectified Diffusion does not experience a much-improved performance from 1 step to 25 steps"
>
> It should be noted that Rectified Diffusion and Rectified Flow are all trained with generated data from pretrained diffusion models. Specifically, in the default setting, they both use 25-step DPMSolver [2] with Stable Diffusion v1-5  to generate noise-data pairs. Therefore, the performance of 25-step DPMSolver with the teacher diffusion model works as the performance upper-bound. The 25-step Rectified Diffusion achieves better performance than  25-step Rectified Flow on both FID and CLIP SCORE. Therefore, Rectified Diffusion has better performance upper bound than Rectified Flow. Additionally, please note that the 25-step performance of Rectified Diffusion and that of  Rectified Flow are already very close to the 25-step DPMSolver with the teacher diffusion model. Therefore, we can not achieve a very large improvement.   **Rectified Diffusion significantly outperforms Rectified Flow in few-step setting.** The performance in few-step setting is the most important criterion to judge the effectiveness of rectification on the ODE (since we aim to achieve efficient sampling and improve the performance in few-step sampling). Rectified Diffusion significantly outperforms Rectified Flow in this setting. Specifically, on COCO-5K. Rectified Flow achieves one-step FID 47.91, two-step FID 31.35. Instead, Rectified Diffusion achieves one-step FID 27.26 (significantly outperforms the two-step FID of Rectified Flow), two-step FID 22.98 (approaching the 25-step FID of Rectified Flow 21.65).
>
>
>
> In simple terms, it is **precisely because our performance in few-step setting significantly surpasses Rectified Flow and closely approaches the performance upper bound** that we do not have the dramatic improvement from 1 to 25 steps, which reflects the superiority of our method Rectified Diffusion.

---

> ### Author Response · Authors · 2024-11-23
> **[2/2]**
>
> **Q3** "Figure 2 is somehow difficult to understand. Why did the FID and CLIP score of Rectified Flow not change with iterations?"
>
> The compared Rectified Flow method [3] did not open source training code. We only have the official weight.   Therefore, we can only get and draw its eventual performance as the baseline. Rectified Diffusion surpasses its eventual performance with only 8% trained images of the official weight of [3].
>
>
>
> **Q4** "Figure 6, the two rightmost plots, why did the curves of Rectified Diffusion go up at 4 and 8 steps? Does it mean the method need to stop at some optimal points before its performance decreases?"
>
> This is a common phenomenon in diffusion models that FID will typically go down and then go up as the CFG [4] value gradually increases.  Many previous works report similar plots (e.g., Figure 9 (b) in [3] and Figure 12 in [5]).
>
> Yes, diffusion models should choose the suitable CFG value to achieve the best performance.
>
>
>
> **Q5** " Figure 7, it seems the biggest difference in preference given by human evaluation is the Phased 4-step results. Does it imply that Phased implementation, which features multiple straight trajectories rather than one single straight trajectory generally, generally gives better results"
>
> We kindly correct you that our target is not to achieve straight trajectories. Straightness is only suitable when we adopt the form of flow-matching or VE.
>
> We rephrase your question into "Figure 7, it seems the biggest difference in preference given by human evaluation is the Phased 4-step results. Does it imply that Phased implementation, which features multiple first-order satisfied trajectories rather than one single first-order satisfied trajectory generally gives better results"
>
>
>
> Yes.  Phased implementation generally gives better implementation since its target ODE typically has a smaller gap with the original ODE.
>
>
>
> **Q6** "Also, in Phased implementation, how many phases were used? It seems this information is not given in the paper."
>
> Four phases are used. This is because PeRFlow [6] adopted four phases for training. We used four phases for training to **set up a fair comparison**.
>
>
>
> [1] Variational Diffusion Models. NeurIPS 2021.
>
> [2] DPM-Solver: A Fast ODE Solver for Diffusion Probabilistic Model Sampling in Around 10 Steps. NeurIPS 2022.
>
> [3] InstaFlow: One Step is Enough for High-Quality Diffusion-Based Text-to-Image Generation. ICLR 2024.
>
> [4] Classifier-Free Diffusion Guidance.
>
> [5] SDXL: Improving Latent Diffusion Models for High-Resolution Image Synthesis. ICLR 2024.
>
> [6] PeRFlow: Piecewise Rectified Flow as Universal Plug-and-Play Accelerator. NeurIPS 2024.

---

> ### Comment · Reviewer_KrPe · 2024-11-26
>
> I have read the rebuttal.

---

> ### Comment · Reviewer_KrPe · 2024-11-26
>
> I have read the rebuttal. I maintain my score.

---

> > ### Author Response · Authors · 2024-11-26
> > **Thank you sincerely for maintaining the positive score!**
> >
> > Thank you so much for your positive review and support—it really means a lot to us. We truly appreciate the time and effort you put into reviewing our work and are happy that our rebuttal helped address your concerns.
> >
> > If you have any other questions, please feel free to reach out.
> >
> > Best regards,
> >
> > The Authors

---

### Official Review · Reviewer_sELj · 2024-11-08

**Soundness:** 3
**Presentation:** 3
**Contribution:** 3
**Rating:** 8
**Confidence:** 5

**Summary:**

This paper proposes Rectified Diffusion that generalizes rectified flow to broader categories of diffusion models, improving image generation efficiency by removing the need for flow-matching and v-prediction. Key components of the proposed method include pre-computed noise-sample pairs, consistency distillation, and phased ODE segmentation, allowing the model to achieve comparable or superior quality with fewer training steps.

**Strengths:**

1. The authors  generalized rectified flow to a wider array of diffusion models by focusing on paired noise-sample training, making the approach compatible with different prediction types and diffusion models.
2. The paper demonstrates that Rectified Diffusion performs better than prior methods like InstaFlow and Rectified Flow in terms of FID and CLIP scores, particularly when generating images in low-step regimes.
3. By dividing the ODE path into phases with enforced linearity in each phase, the model maintains a balance between quality and computational efficiency without altering the fundamental diffusion trajectory.

**Weaknesses:**

1.  While the approach seems efficient, the paper builds on existing methods, particularly using paired noise-sample training, phased ODE segmentation and consistency distillation. The lack of entirely new techniques could limit its novelty in diffusion model research.
2. While this method employs distillation techniques that improve inference speed, achieving single-step image generation without pre-trained models could introduce a novel advancement.
3. The performance metrics on FID shows only marginal improvements over Rectified Flow. The FID metric for Rectified Flow shown in Table 2 corresponding to step2 (62.81) is higher than step1 (13.67). Are these values correct? Generally step2 value should be lower.
4. Although the method is compared against multiple models, all experiments are limited to a single dataset (COCO). Additional experiments on other benchmark datasets, such as ImageNet, would help validate the generalizability of the results.

**Questions:**

1. The performance metrics on FID shows only marginal improvements over Rectified Flow. The FID metric for Rectified Flow shown in Table 2 corresponding to step2 (62.81) is higher than step1 (13.67). Are these values correct? Generally step2 value should be lower.
2. Could the authors validated the performance of their method on more datasets?

---

> ### Author Response · Authors · 2024-11-23
> **[1/3]**
>
> Thank you for the in-depth review.
>
>
>
> **W1** "While the approach seems efficient, the paper builds on existing methods, particularly using paired noise-sample training, phased ODE segmentation, and consistency distillation. The lack of entirely new techniques could limit its novelty in diffusion model research."
>
>
>
> The motivation for Rectified Diffusion is to investigate and rethink the most essential part of rectified flow. The investigation allows us to extend the scope of rectification, which was originally proposed and believed only suitable for rectified flow (using the form $\mathbf x_t = t \mathbf x_0 + (1-t) \boldsymbol \epsilon$, to general diffusion models.
>
> In particular, InstaFlow [1], an important follow-up work to rectified flow that extends it to text-to-image tasks and serves as a key baseline in our paper, uses the Stable Diffusion (Which is an epsilon prediction neural network follows DDPM diffusion form) for initialization. However, in InstaFlow, the model is first converted into a v-prediction flow-matching model before undergoing retraining of paired noise-sample This process introduces a gap between the pretrained model and the retrained model supporting efficient sampling.
>
> The conversion that converts the pretrained diffusion models (can be forms of DDPM, Sub-VP, VE) has become a common practice in the important following works [1,2,3].
>
> We are the first to point out that this conversion is unimportant and show that straightness is not the essential training target. Any ODE (even curved) can achieve one-step sampling as long as it satisfies the requirement of first-order property.
>
> Therefore, our main contribution is to rethink and break the incomplete understanding from previous research instead of proposing new techniques, new network designs, optimizer configurations, etc.
>
> Additionally, building upon existing methods makes our method as simple, straightforward, and accessible as possible to most diffusion-related researchers.  Essentially, Rectified Diffusion does not need to change anything of pretrained diffusion models for sampling acceleration (including networks, preconds, training/inference code, noise scheduler, etc.). The only difference as highlighted in our paper is to replace the noise and data in standard diffusion training with paired noise (precollected) and data (generated).
>
> With this simple design, we still achieve significant improvement compared to the previous best-performing rectified flow-based methods (with even smaller training costs) and comparable performance to previous best-performing distillation or GAN-based acceleration methods.
>
> We believe this discovery can inspire related research and broaden the understanding of diffusion models in the community.
>
>
>
>
>
> **W2** "While this method employs distillation techniques that improve inference speed, achieving single-step image generation without pre-trained models could introduce a novel advancement."
>
> Rectified Flow-based methods and Rectified Diffusion all require pretrained models to collect paired noise and data. We cannot collect such pairs without pretrained models.

---

> > ### Comment · Reviewer_sELj · 2024-11-25
> >
> > Agreed to your detailed explanation

---

> ### Author Response · Authors · 2024-11-23
> **[2/3]**
>
> **W3.1** "FID improvements over Rectified Flow."
>
>
>
> **Rectified Diffusion has better performance upper bound than Rectified Flow.** It should be noted that Rectified Diffusion and Rectified Flow are all trained with generated data from pretrained diffusion models. Specifically, in the default setting, they both use 25-step DPMSolver [4] with Stable Diffusion v1-5  to generate noise-data pairs. Therefore, the performance of 25-step DPMSolver with the teacher diffusion model works as the performance upper-bound. The 25-step Rectified Diffusion achieves better performance than  25-step Rectified Flow on both FID and CLIP SCORE. Therefore, Rectified Diffusion has better performance upper bound than Rectified Flow. Additionally, please note that the 25-step performance of Rectified Diffusion and that of Rectified Flow are already very close to the 25-step DPMSolver with the teacher diffusion model. Therefore, we can not achieve a very large improvement.
>
>
>
> **Rectified Diffusion significantly outperforms Rectified Flow in few-step setting.** The performance in few-step setting is the most important criterion to judge the effectiveness of rectification on the ODE (since we aim to achieve efficient sampling and improve the performance in few-step sampling). Rectified Diffusion significantly outperforms Rectified Flow in this setting. Specifically, on COCO-5K. Rectified Flow achieves one-step FID 47.91, and two-step FID 31.35. Instead, Rectified Diffusion achieves one-step FID 27.26 (significantly outperforms the two-step FID of Rectified Flow), two-step FID 22.98 (approaching the 25-step FID of Rectified Flow 21.65). The is significant evidence that Rectified Diffusion is better than Rectified Flow.
>
>
>
> **Rectified Diffusion outperforms Rectified Flow with significantly smaller training costs.**
>
> Please kindly refer to Figure 2, where we show the performance change as the training iterations. Our one-step performance significantly outperforms the one-step performance of the Rectified Flow official weight within only 20, 000 iterations with batch size 128. The overall number of trained images at this iteration is only 8% of the number of trained images in Rectified Flow (lines 498-499, lines 123-125).
>
> Therefore, we believe our overall empirical improvement is **considerably significant instead of marginal**.
>
>
>
> **W3.2** "The FID metric for Rectified Flow shown in Table 2 corresponding to step2 (62.81) is higher than step1 (13.67). Are these values correct? Generally step2 value should be lower."
>
> Yes. Our reported performance is correct and was tested with the official weight. Rectified Flow (Distill) is the rectified flow model further trained through naive distillation after rectification. It lacks the ability of multistep refinement (as stated in lines 507-508). This is also consistent with the performance reported in Table 1. (lines 397-399).
>
> Understanding this drawback of Rectified Flow (Distill), we instead choose consistency distillation (as stated in lines 376-380) to further improve performance. We achieve better one-step performance with only 3% training time of Rectified Flow (Distill) and support multistep sampling to further improve the performance (lines 493-494).

---

> ### Author Response · Authors · 2024-11-23
> **[3/3]**
>
> **W4** "Although the method is compared against multiple models, all experiments are limited to a single dataset (COCO). Additional experiments on other benchmark datasets, such as ImageNet, would help validate the generalizability of the results."
>
> Thank you for your suggestion. We additionally evaluate model performance on the Laion and CC3M. We follow the test setting of COCO-2017, adopting 5k subset for evaluation. Our experimental results show that Rectified Diffusion still consistently surpasses Rectified Flow, which is consistent with our original evaluation on the COCO. We believe this is strong evidence of the superiority of Rectified Diffusion.
>
> Laion Subset
>
> | Step  | Metric         | Rectified Diffusion             | Rectified Flow |
> | ----- | -------------- | ------------------------------- | -------------- |
> | **8** | **CLIP Score** | 26.36                           | 25.49          |
> |       | **FID**        | 23.52                           | 24.73          |
> | **4** | **CLIP Score** | 25.80                           | 24.65          |
> |       | **FID**        | 24.70                           | 27.27          |
> | **2** | **CLIP Score** | 25.15                           | 23.38          |
> |       | **FID**        | 26.14                           | 35.09          |
> | **1** | **CLIP Score** | **23.30 ( $\uparrow 3.18$)**    | **20.12**      |
> |       | **FID**        | **30.14 ( $\downarrow 22.72$)** | **52.86**      |
>
> CC3M Subset
>
> | Step  | Metric         | Rectified Diffusion              | Rectified Flow |
> | ----- | -------------- | -------------------------------- | -------------- |
> | **8** | **CLIP Score** | 28.33                            | 27.65          |
> |       | **FID**        | 28.10                            | 29.07          |
> | **4** | **CLIP Score** | 28.08                            | 27.37          |
> |       | **FID**        | 29.97                            | 31.54          |
> | **2** | **CLIP Score** | 27.68                            | 26.27          |
> |       | **FID**        | 31.56                            | 37.72          |
> | **1** | **CLIP Score** | **26.30 ( $\uparrow 1.67$)**     | **24.63**      |
> |       | **FID**        | **34.28  ( $\downarrow 14.78$)** | **49.06**      |
>
> **Q1** "The performance metrics on FID show only marginal improvements over Rectified Flow. The FID metric for Rectified Flow shown in Table 2 corresponding to step2 (62.81) is higher than step1 (13.67). Are these values correct? Generally, step2 value should be lower."
>
> Please refer to W3.2.
>
> **Q2** "Could the authors validate the performance of their method on more datasets?"
>
> Please refer to W4.
>
> [1] InstaFlow: One Step is Enough for High-Quality Diffusion-Based Text-to-Image Generation. ICLR 2024.
>
> [2] Improving the Training of Rectified Flows. NeurIPS 2024.
>
> [3] PeRFlow: Piecewise Rectified Flow as Universal Plug-and-Play Accelerator. NeurIPS 2024.
>
> [4] DPM-Solver: A Fast ODE Solver for Diffusion Probabilistic Model Sampling in Around 10 Steps. NeurIPS 2022.

---

> > ### Comment · Reviewer_sELj · 2024-11-25
> >
> > Thank you for sharing your results on additional datasets

---

> > > ### Author Response · Authors · 2024-11-26
> > > **Thank you sincerely for the positive review and support!**
> > >
> > > Thank you sincerely for the positive review and support! We deeply appreciate the time and effort you invested in reviewing our work and are glad our rebuttal resolved your concerns.
> > >
> > > Please feel free to reach out if you have any questions.
> > >
> > > Best regards,
> > >
> > > The Authors

---

### Author Response · Authors · 2024-11-25
**General response**

We are sincerely grateful to the reviewers for dedicating their time and effort to review our work. We are delighted to see reviewers commenting on our work with "convincing and superior performance", "original yet simple".

In this rebuttal, we have given careful thought to the reviewers’ suggestions and made the following revisions to our manuscript to answer the questions and concerns:

- In **Figure 1**, we revise the figure to better present the core motivation and advantages of Rectified Diffusion: Simpler, Stronger, and General.

- In **Figure 2**, we enrich the caption to show that the dashed lines are obtained from the official weight of Rectified Flow. This better showcases that our method can significantly surpasses the final performance of Rectified Flow with only 8% trained images and within 10% total iterations trained.

- In **Figure 4 and Figure 5**, we highlight that  "Since it’s hard to visually tell whether a curved ODE path satisfies first-order
property, we apply straight lines for more clear demonstration." to avoid any confusion as we originally stated in lines 292-294.

- In **Figure 6**, we enrich the captions to make it more accessible.

- In **Equation 1 and Theorem 1**, we add additional introduction to the basic property of $\lambda_t = \ln \frac{\alpha_t}{\sigma_t}$ to make our paper more self-contained and friendly to readers.

- In the **Supplementary Section 3**, we provide further elaboration on why perfect coupling will lead to first-order ODE.

- In the **Supplementary Section 4**, we provide our validation summary and additional validation results on more datasets.

- In the **Supplementary Section 5.1**, we provide a concise summary of our motivation and contribution, aiming to clarify the research context, highlight the challenges addressed, and emphasize the significance of our findings in advancing the field.
-  In the **Supplementary Section 5.2**, we provide a more detailed clarification on the difference between rectified diffusion and consistency distillations.
- In the  **Supplementary Section 5.3**, we provide a clarification on the effectiveness of the additional distillation procedure after rectification.


We have highlighted the revised part in our manuscript in blue colour.

Please refer to the following rebuttals for other specific concerns and more details. We are looking forward to your further reply and discussion. Please feel free to share any additional questions or feedback, and we’ll be happy to provide further clarification.

Sincerely,

The Authors

---

### Meta-Review · Area_Chair_JJUE · 2024-12-19

**Metareview:**

This submission aims to generalise rectification to a wider class of diffusion models via first-order ODEs. Authors propose that precomputed noise-data pairs constitute the essential component (c.f. flow-matching, v-prediction), to enable a learned diffusion ODE to be closer to a first-order ODE. Experimental work can evidence improved image generation efficiency; comparable or superior quality with fewer training steps (c.f. rectified flow, phased ODE segmentation).

AC notes that there are interesting ideas in the paper. The writing quality could be improved, e.g. the way theorems are stated. Reviews allude to questions regarding results supporting claims. A further important point to note is that non-straight interpolating curves are not new. Previous work from the last two years has investigated other approaches to straightening integration paths [1,2]. These works are not mentioned or compared, however they were shown to improve flow-matching models and report experiments with non-linear paths. Both are well cited and so may not be considered obscure.

This paper was discussed at length with the SAC.

After reviewing the paper, rebuttal and resulting discussions we find that the overall strengths outweigh the weaknesses and recommends acceptance. For the camera-ready version, the authors should incorporate all key results presented in the rebuttal. In particular, the review process highlighted the need to improve clarity and understanding surrounding straight, curved ODE-trajectories, first-order linearity and acknowledgement of key missing works relating to the straightening of integration paths.

1. Tong et al. Improving and generalizing flow-based generative models with minibatch optimal transport
2. Albergo et al. Stochastic Interpolants: A Unifying Framework for Flows and Diffusions

**Additional Comments On Reviewer Discussion:**

The paper received four reviews resulting in divergent scores: two clear accepts, one borderline accept and one strong reject.

Positive reviewers noted the originality and simplicity of the core idea. Multiple reviewers recognised strong quantitative and qualitative results. The author rebuttal could successfully alleviate a subset of concerns regarding: (i) misunderstandings surrounding straight, curved ODE-trajectories and first-order linearity; (ii) significance of empirical improvements; (iii) additional benchmarks and performance upper bounds; (iv) self-consistency constraints.

The strongly negative reviewer raised further concerns relating to (v) motivation for the investigation; (vi) theorem novelty; and latterly (vii) approximation errors and constant assumptions. Authors attempt to address these concerns through additional explanation however the reviewer remained unconvinced.

---

### Decision · Program_Chairs · 2025-01-22

Accept (Poster)